# Learning Tight Rejection Boundaries without Negatives for Strict One-Class Audio Deepfake Detection

**Yuze Zhao**[1]   **Kuiyuan Zhang**[2]   **Zhongyun Hua**[1]   **Yushu Zhang**[3]   **Qing Liao**[1]   **Wei Jiang**[2]

## Abstract

The rapid evolution of audio deepfakes requires robust detection capable of generalizing to unseen attacks. One-class learning offers inherent robustness for this task by characterizing real speech distributions to detect anomalies. However, establishing a compact decision boundary without spoof supervision remains a fundamental challenge. Existing "relaxed" approaches often compromise this strictness by introducing auxiliary negative samples, which biases the boundary toward seen artifacts and degrades generalization to unseen attacks. To address this, we propose CA-SOADD, a framework that refines the acceptance region by constructing off-manifold boundary probes. Our proposed centroid-anchored tri-objective learning paradigm simultaneously enforces centroid compactness and a centroid-referenced margin against these probes, thereby explicitly tightening the acceptance region without treating them as an explicit negative class. We further extend the framework to heterogeneous settings through domain-conditioned centroids. Experiments on ASVSpoof, CtrSVDD and MLAAD benchmarks demonstrate that our strict real-only method consistently outperforms strong baselines under unseen attack types and domain shifts, with its effectiveness further validated through extensive ablation studies.

[1]Harbin Institute of Technology, Shenzhen, Guangdong 518055, China [2]School of Cyberspace Science, Harbin Institute of Technology, Harbin 150001, China [3]School of Computing and Artificial Intelligence, Jiangxi University of Finance and Economics, Nanchang 330013, China. Correspondence to: Wei Jiang <jw@hit.edu.cn>.

*Proceedings of the $43^{rd}$ International Conference on Machine Learning*, Seoul, South Korea. PMLR 306, 2026. Copyright 2026 by the author(s).

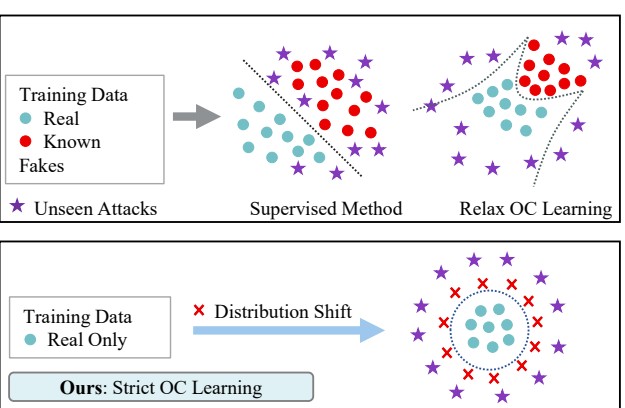

*Figure 1.* Comparison of three training paradigms for audio deepfake detection: supervised detection, relaxed OCL, and strict OCL.

## 1. Introduction

Modern speech generation systems, such as text-to-speech (TTS) (Ju et al., 2024; Wang et al., 2025) and voice conversion (VC) (Wang et al., 2022; Ma et al., 2024), can now produce highly realistic synthetic speech at scale, introducing serious security risks such as impersonation and fraud. Although supervised audio deepfake detectors perform well on seen attacks, they often overfit to generator-specific artifacts (Jung et al., 2022; Tak et al., 2021; Jung et al., 2020), degrading significantly against unseen or evolving attacks. To bypass this reliance on specific spoof patterns, **one-class learning (OCL)** models the bona fide distribution exclusively, treating deviations as anomalies.

A common strict OCL strategy enforces bona fide compactness in the embedding space, e.g., by pulling embeddings toward a center or shrinking the hypersphere volume (Schölkopf et al., 2001; Ruff et al., 2018). However, compactness alone does not ensure a tight rejection boundary. Under strict OCL, the acceptance region around the center may remain insufficiently constrained, allowing diverse spoof samples to drift into the centroid neighborhood. This weakness becomes even more pronounced under benchmark shifts and high-fidelity attacks (Zhang et al., 2025).

To explicitly shape boundaries, many previous works relax the OCL formulation by introducing auxiliary spoof samples during training (Zhang et al., 2021; Kim et al., 2024; Ye et al., 2025). Although effective within benchmark set-

tings, this practice can bias the learned boundary toward the distribution of the seen spoofs. As a result, the model primarily learns to reject specific negative families, which may lead to poor generalization when test-time attacks differ from these auxiliary samples. In open-world deepfake detection, where spoof pipelines evolve continuously, such dependence undermines the motivation for strict OCL.

An alternative approach is to generate distribution-shifted views from bona fide speech and treat them as explicit negatives (e.g., via BCE or contrastive discrimination). However, this risks reducing strict OCL to a form of surrogate discrimination, where the decision boundary becomes overfitted to the specific shift types introduced during training. Moreover, standard OCL inference typically relies on simple scoring rules, such as cosine similarity to a centroid. Training a discriminative classifier against synthetic negatives creates an objective-inference mismatch, as the training objective no longer aligns with the inference metric.

We address the boundary limitation of strict OCL through **off-manifold boundary probes**. Rather than defining an explicit negative class, these distribution-shifted views impose centroid-referenced margin constraints that directly delineate the acceptance region under similarity scoring. For each bona fide utterance, we generate a structure-disrupting view as an off-manifold probe. Training ensures that bona fide embeddings remain significantly closer to the running centroid than their corresponding probes by a predefined margin. This mechanism effectively compresses the decision volume, tightening the boundary without relying on spoof data or over-fitting to generator-specific artifacts.

Concretely, we realize this framework through **Centroid-Anchored Tri-Objective Learning**, which consists of three real-only objectives defined around a running centroid. $OBJ_{cpt}$ **(centroid compactness)** draws bona fide embeddings toward the centroid to establish a tight genuine core. $OBJ_{binv}$ **(benign invariance)** preserves representation stability under benign perturbations, preventing nuisance variations from distorting the centroid neighborhood. $OBJ_{cabs}$ **(centroid-anchored boundary shaping)** employs structure-disrupting, distribution-shifted views solely as off-manifold boundary probes through a centroid-referenced margin, without introducing an explicit negative class.

Based on this principle, we propose **CA-SOADD** (Centroid-Anchored Strict One-Class Audio Deepfake Detection), a real-only framework for audio deepfake detection. Strictly following a bona fide-only protocol, we exclude spoof data and labels from training, validation, model selection, and threshold calibration. Our contributions are summarized as follows: (i) We introduce centroid-referenced margin constraints using probe views to shape the acceptance region under cosine-to-centroid scoring in a strict real-only setting. (ii) We propose a tri-objective, centroid-anchored

training paradigm that enforces compactness, benign invariance, and a boundary margin. We also introduce a bona fide-defined running reference-frame alignment module to mitigate the *cosine-collapse* issue and increase the effective angular range for centroid-based scoring. (iii) We extend the method to heterogeneous scenarios through domain-conditioned centroids, and evaluate it under strict real-only protocols with cross-domain transfer and ablations.

## 2. Related Work

Recent progress in speech generation has increased both the realism and diversity of audio deepfakes. However, most state-of-the-art detectors remain reliant on supervised training with spoof data, making them prone to overfitting artifacts from seen generation pipelines. Furthermore, many one-class formulations used in practice still rely on spoof or auxiliary negatives for training or calibration, rendering them incompatible with strict real-only deployment. We therefore revisit one-class learning utilizing exclusively bona fide speech, focusing on the central challenge: establishing a deployable decision boundary without negatives.

### 2.1. One-Class Learning and the Boundary Problem

One-class learning models the distribution of bona fide speech and treats deviations as anomalies. OC-SVM (Schölkopf et al., 2001) learns a maximum-margin boundary that encloses bona fide embeddings. Deep-SVDD (Ruff et al., 2018) learns a representation that minimizes the volume of a hypersphere centered at a reference point. In audio anti-spoofing, OC-Softmax introduces a margin-based one-class softmax objective but explicitly utilizes spoof samples as negatives (Zhang et al., 2021); ACS updates the bona fide centroid online (Kim et al., 2024), while KLOC-Softmax further regularizes the embedding geometry to refine one-class separation (Ye et al., 2025). Collectively, these methods highlight a core difficulty: relying solely on real-only compactness does not uniquely specify a tight, deployable rejection boundary.

In practice, many OCL pipelines adopt a *relaxed* formulation by introducing auxiliary spoof/negative samples during training, or by using spoof-guided validation and calibration. Such practices can induce spoof-distribution-dependent boundaries, biasing the detector toward seen artifact families while leaving other deviation modes weakly constrained.

Our work targets this challenge by explicitly shaping a rejection boundary aligned with the centroid-based inference score, rather than relying solely on bona fide clustering.

### 2.2. Distribution-Shifted Views as Boundary Probing

A closely related line of research refines novelty boundaries using synthetic outliers derived from in-distribution

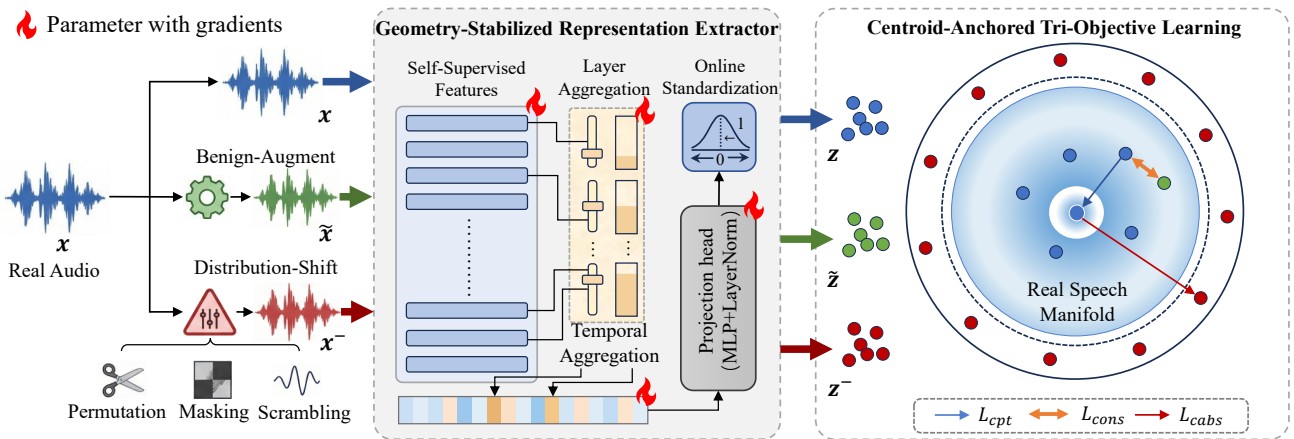

*Figure 2.* Overview of the CA-SOADD pipeline. We illustrate the structure-disrupting operators for generating distribution-shifted views and the embedding intuition: benign augmentations keep $\tilde{z}$ aligned with $z$, while the shifted $z^-$ acts as an off-manifold boundary probe.

data. For instance, CSI (Tack et al., 2020) treats distributionally shifted instances as out-of-distribution surrogates via contrastive learning, and NAD (Sinha et al., 2021) designs procedures to generate hard negatives from in-distribution samples for data augmentation. In computer vision, pseudo anomaly generation methods like CutPaste (Li et al., 2021) demonstrate that carefully crafted synthetic anomalies enhance novelty detection. While these studies establish the utility of shifted views as boundary-shaping signals, their training objectives often align with surrogate discrimination (e.g., binary classification) rather than strict, centroid-scored one-class detection.

Our method differs fundamentally by avoiding surrogate discrimination: we treat distribution-shifted views solely as off-manifold probes to enforce a **centroid-referenced margin**, rather than as an explicit negative class. This formulation ensures strict alignment with a cosine-to-centroid scoring rule, allowing the model to shape a tight decision boundary without using external negatives or spoof data.

## 3. Methodology

### 3.1. Overview

We study audio deepfake detection under **strict one-class learning**, where training uses *only* bona fide speech. CA-SOADD follows a **score–loss alignment** principle: inference relies on a *single* cosine-to-centroid score, and training shapes the boundary exclusively through *centroid-referenced margin constraints*, rather than learning a classifier with an explicit negative class. We do not optimize a discriminator between $x$ and $x^-$; shifted views only enter the objective to enforce margins relative to the centroid.

As illustrated in Fig. 2, CA-SOADD comprises a **Geometry-Stabilized Representation Extractor** that produces utterance embeddings optimized for centroid similarity, and a **Centroid-Anchored Tri-Objective Learning** scheme that

learns a compact bona fide manifold and a calibrated rejection boundary from genuine data alone.

**Inputs and three objectives.** Given a bona fide utterance $x$, the extractor $f_\theta$ processes three views, each tied to a complementary objective. (i) **OBJ$_{cpt}$ (centroid compactness):** the embedding $z = f_\theta(x)$ is pulled toward a running bona fide centroid $c$ to condense the bona fide core. (ii) **OBJ$_{binv}$ (benign invariance):** a benignly augmented view $\tilde{x} = \mathcal{T}(x)$ yields $\tilde{z} = f_\theta(\tilde{x})$, ensuring representation stability within the centroid neighborhood against nuisance perturbations. (iii) **OBJ$_{cabs}$ (centroid-anchored boundary shaping):** a structure-disrupting transformation $\mathcal{A}(\cdot)$ (e.g., permutation, masking, scrambling) generates a distribution-shifted view $x^- = \mathcal{A}(x)$ with embedding $z^- = f_\theta(x^-)$. $\mathcal{A}$ is defined independently of spoof data and kept unchanged across datasets. $x^-$ is used *only as an off-manifold boundary probe* via a centroid-referenced margin, rather than as an explicit negative in surrogate discrimination.

**Training and inference.** We maintain the bona fide centroid $c$ as an online statistic updated solely from bona fide embeddings. The model jointly minimizes the three losses in Sec. 3.3; at inference, the detection score is the cosine similarity to global centroid $c$ (or domain-specific centroid $c_d$ in heterogeneous settings): $\text{sim}(z, c) = z^\top c / (\|z\|_2 \|c\|_2)$.

### 3.2. Geometry-Stabilized Representation Extractor

Given an input waveform $x$, the geometry-stabilized representation extractor $f_\theta$ produces an utterance embedding $z \in \mathbb{R}^D$ for centroid-based strict one-class scoring. Built on a pretrained self-supervised speech encoder, it is designed to stabilize embedding geometry, ensuring reliable cosine-to-centroid scoring under real-only training. Concretely, it performs a lightweight layer–time selection to aggregate multi-layer frame-level features, maps the result with a projection head, and applies bona fide–only online standardization to reduce scale and distribution drift that

would otherwise destabilize centroid statistics.

### 3.2.1. HIERARCHICAL FEATURE AGGREGATION

To obtain robust utterance-level representations, we aggregate features along two axes: network depth (layers) and time. Layer aggregation mitigates layer-specific variability in pretrained encoders. Temporal aggregation emphasizes informative frames and suppresses nuisance segments, improving centroid stability.

**Layer Aggregation.** Let $\{\boldsymbol{H}^{(l)}\}_{l=1}^{L}$ denote the hidden states from the $L$ Transformer layers of the speech encoder, where $\boldsymbol{H}^{(l)} \in \mathbb{R}^{T \times d}$. We introduce a learnable parameter vector $\boldsymbol{s} \in \mathbb{R}^{L}$ to compute normalized layer weights $\boldsymbol{\alpha}$ via a temperature-scaled softmax:

$$\boldsymbol{\alpha} = \text{softmax}(\boldsymbol{s}/\tau) \in \mathbb{R}^{L}, \tag{1}$$

where $\tau$ controls sharpness. The layer-aggregated features are computed as a weighted sum: $\boldsymbol{F} = \sum_{l=1}^{L} \boldsymbol{\alpha}_l \boldsymbol{H}^{(l)} \in \mathbb{R}^{T \times d}$. To prevent collapse to a single layer, we optionally minimize the negative entropy $\mathcal{L}_{\text{ent}} = \sum \boldsymbol{\alpha}_l \log(\boldsymbol{\alpha}_l + \epsilon)$, maximizing the diversity of layer utilization.

**Temporal Aggregation.** We then pool the temporal sequence $\boldsymbol{F} \in \mathbb{R}^{T \times d}$ into a single vector using a lightweight attention module. Unnormalized attention scores $\boldsymbol{a} \in \mathbb{R}^{T}$ are generated via a two-layer bottleneck network $\phi(\cdot)$ comprising $1 \times 1$ convolutions:

$$\boldsymbol{a} = \phi(\boldsymbol{F}) = \text{Conv}_{h \to 1}(\tanh(\text{Conv}_{d \to h}(\boldsymbol{F}))). \tag{2}$$

Normalized temporal weights $\boldsymbol{\beta} \in \mathbb{R}^{T}$ are obtained via softmax, $\beta_t = \exp(a_t)/\sum_j \exp(a_j)$. These weights produce an attentive pooling result:

$$\boldsymbol{r} = \sum_{t=1}^{T} \beta_t \boldsymbol{F}_t \in \mathbb{R}^{d}. \tag{3}$$

This lightweight depth–time aggregation yields stable utterance embeddings that support reliable centroid estimation under strict real-only training. Finally, we map $\boldsymbol{r}$ to the target embedding space with a learnable projection head $g(\cdot)$ to obtain $\boldsymbol{z} = g(\boldsymbol{r}) \in \mathbb{R}^{D}$. We implement $g(\cdot)$ as a lightweight 2-layer MLP with LayerNorm and ReLU.

### 3.2.2. ONLINE STANDARDIZATION

Strict one-class training can exhibit a *cosine-collapse* pathology, where bona fide embeddings become overly concentrated (near-saturated pairwise cosine similarity), leaving little effective angular dynamic range for centroid-based scoring and boundary shaping. To mitigate this, we align embeddings to a bona fide-defined running reference frame by standardizing them with running statistics estimated solely from bona fide samples.

Specifically, we maintain running statistics (mean $\boldsymbol{\mu}$ and variance $\boldsymbol{\sigma}^2$) updated via a Welford-style online estimator using bona fide mini-batches (without gradient propagation). The standardized embedding $\boldsymbol{z}$ is obtained as:

$$\boldsymbol{z} = \frac{\boldsymbol{z} - \boldsymbol{\mu}}{\sqrt{\boldsymbol{\sigma}^2 + \epsilon}}, \tag{4}$$

The $(\boldsymbol{\mu}, \boldsymbol{\sigma}^2)$ are updated during training and fixed at inference, and the pseudo-code is provided in Appendix B.

### 3.3. Centroid-Anchored Tri-Objective Learning

We train CA-SOADD with centroid-anchored tri-objective learning under a strict real-only setting (Fig. 2), aiming to learn a compact bona fide manifold and a deployable decision boundary *without* spoof data. Training is anchored at a running bona fide embedding centroid, and jointly optimizes centroid compactness (OBJ$_{\text{cpt}}$), benign-view invariance (OBJ$_{\text{binv}}$), and centroid-anchored boundary shaping with distribution-shifted views (OBJ$_{\text{cabs}}$).

### 3.3.1. OBJ$_{\text{CPT}}$: CENTROID COMPACTNESS

We maintain a global bona fide centroid $\boldsymbol{c} \in \mathbb{R}^{D}$ as a non-learnable reference. To ensure the reference stably converges to the population mean of genuine speech, $\boldsymbol{c}$ is updated via cumulative moving average (CMA) using the batch mean $\boldsymbol{\mu}_{\text{batch}}$:

$$\boldsymbol{c} \leftarrow \frac{N \boldsymbol{c} + B \boldsymbol{\mu}_{\text{batch}}}{N + B}, \qquad N \leftarrow N + B, \tag{5}$$

where $N$ is the total number of accumulated bona fide samples and $B$ is the batch size. We then encourage compactness by maximizing cosine similarity:

$$\mathcal{L}_{\text{cpt}} = 1 - \frac{1}{B} \sum_{i=1}^{B} \text{sim}(\boldsymbol{z}_i, \boldsymbol{c}). \tag{6}$$

### 3.3.2. OBJ$_{\text{BINV}}$: BENIGN INVARIANCE

To stabilize representations against nuisance variability (e.g., channel and noise), we generate a benign augmentation $\tilde{\boldsymbol{x}} = \mathcal{T}(\boldsymbol{x})$ (detailed in Appendix C.4) and align the augmented embedding with the clean one: $\mathcal{L}_{\text{binv}} = 1 - \text{sim}(\boldsymbol{z}, f_\theta(\tilde{\boldsymbol{x}}))$. This regularizer reduces benign-induced drift near the centroid, which is crucial under cosine-to-centroid inference.

### 3.3.3. OBJ$_{\text{CABS}}$: BOUNDARY SHAPING

OBJ$_{\text{cpt}}$ and OBJ$_{\text{binv}}$ constrain local bona fide structure but do not explicitly control nearby off-manifold regions. To shape a rejection boundary without spoof data, we generate a distribution-shifted view $\boldsymbol{x}^- = \mathcal{A}(\boldsymbol{x})$ using structure-disrupting transformations, compute $\boldsymbol{z}^- = f_\theta(\boldsymbol{x}^-)$, and

impose a centroid-referenced margin constraint:

$$\mathcal{L}_{\text{cabs}} = \max\left(0,\; m - \text{sim}(\boldsymbol{z}, \boldsymbol{c}) + \text{sim}(\boldsymbol{z}^-, \boldsymbol{c})\right), \quad (7)$$

where $m > 0$ is a scalar margin. This enforces $\text{sim}(\boldsymbol{z}, \boldsymbol{c}) \geq \text{sim}(\boldsymbol{z}^-, \boldsymbol{c}) + m$, tightening the acceptance region under the same cosine-to-centroid scoring rule used at inference. We set $m = 1.0$ as the default margin in all main experiments and provide a sensitivity analysis in Appendix F.2.

Crucially, $\boldsymbol{x}^-$ is not treated as an explicit negative class: OBJ$_{\text{cabs}}$ does not optimize a discriminator between $\boldsymbol{x}$ and $\boldsymbol{x}^-$. Using shifted views as supervised negatives (e.g., BCE) would shape the boundary around the chosen shift distribution, whereas centroid-referenced margins use them only as boundary probes aligned with centroid scoring. The transformation family $\mathcal{A}$ is detailed in Appendix C.

### 3.3.4. OVERALL OBJECTIVE

We jointly optimize the three proposed components along with the entropy regularization term:

$$\mathcal{L} = \lambda_1 \mathcal{L}_{\text{cpt}} + \lambda_2 \mathcal{L}_{\text{binv}} + \lambda_3 \mathcal{L}_{\text{cabs}} + \lambda_4 \mathcal{L}_{\text{ent}}, \quad (8)$$

where $\lambda(\cdot)$ are non-negative weights. All augmentations and distribution shifts are applied on-the-fly during training.

### 3.4. Multi-Domain Extension

Real-world bona fide speech often exhibits heterogeneity across languages, channels, or recording devices, inducing a multi-domain distribution that a single global centroid may fail to capture. To account for such domain shifts while preserving strict one-class constraints, we extend our framework with *domain-conditioned centroids*.

**Shared Extractor with Domain-Specific Statistics.** Rather than training separate models, we employ a single shared extractor $f_\theta$ for all data. We address domain shifts by maintaining a set of domain-specific running centroids $\{\boldsymbol{c}_\ell\}_{\ell=1}^K$ and learnable layer-selection parameters $\{\boldsymbol{s}_\ell\}_{\ell=1}^K$. This yields a lightweight extension where the backbone and projection head remain shared, while the reference centroid and feature aggregation weights are conditioned on the domain label $\ell$.

**Centroid Updates.** Each training utterance $\boldsymbol{x}_i$ is associated with a domain label $\ell_i$. We maintain a running centroid $\boldsymbol{c}_\ell$ and sample count $N_\ell$ for each domain. Given a mini-batch, let $\mathcal{B}^{(\ell)}$ denote the set of bona fide embeddings belonging to domain $\ell$, with size $B_\ell = |\mathcal{B}^{(\ell)}|$. We update the corresponding centroid via the CMA:

$$\boldsymbol{c}_\ell \leftarrow \frac{N_\ell \boldsymbol{c}_\ell + B_\ell \boldsymbol{\mu}_\ell}{N_\ell + B_\ell}, \qquad N_\ell \leftarrow N_\ell + B_\ell, \quad (9)$$

where $\boldsymbol{\mu}_\ell$ is the batch mean of $\mathcal{B}^{(\ell)}$. If $B_\ell = 0$, the centroid remains unchanged.

**Domain-Conditioned Objectives.** The training objectives are adapted as follows. i) for OBJ$_{\text{cpt}}$, we maximize cosine similarity between the embedding $\boldsymbol{z}$ and its corresponding ground-truth domain centroid $\boldsymbol{c}_\ell$ for the input sample with a domain label $\ell$. ii) for OBJ$_{\text{cabs}}$, to ensure distribution-shifted views $\boldsymbol{z}^-$ are pushed away from the entire bona fide manifold, we define the margin constraint against the nearest centroid:

$$\mathcal{L}'_{\text{cabs}} = \max\left(0, m - \text{sim}(\boldsymbol{z}, \boldsymbol{c}_\ell) + \max_k \text{sim}(\boldsymbol{z}^-, \boldsymbol{c}_k)\right). \quad (10)$$

This modification ensures that off-manifold probes are discouraged from occupying any region of the bona fide space, regardless of the domain.

### 3.5. Inference

At inference time, we assume the domain label $\ell$ (e.g., language) is known. Given a test utterance $\boldsymbol{x}$, we compute its embedding $\boldsymbol{z} = f_\theta(\boldsymbol{x})$ and define the anomaly score as the cosine similarity to the corresponding domain centroid:

$$\text{score}(\boldsymbol{x}) = \text{sim}(\boldsymbol{z}, \boldsymbol{c}_\ell). \quad (11)$$

A higher score indicates higher bona fide confidence.

## 4. Experimental Setting

### 4.1. Datasets and Protocols

We evaluate CA-SOADD on the ASVSpoof benchmarks, the CtrSVDD singing voice benchmark, and the MLAAD cross-lingual benchmark.

**Strict real-only protocol.** We follow a strict real-only protocol throughout: no spoofed utterances and no spoof labels are used for training, validation, model selection, or threshold calibration. All models are trained on bona fide speech only. We select checkpoints using a real-only validation set by minimizing the recipe-specific total loss.

**Scoring and evaluation.** We use cosine-to-centroid similarity for inference and report AUC and EER as standard metrics in ASVSpoof-style evaluation. To reflect realistic real-only deployment scenarios where spoof data are unavailable for calibration, we additionally perform real-only threshold calibration using only the score distribution of a bona fide development split. Specifically, we choose operating points by controlling the target false-reject rate on bona fide trials, and then report the corresponding false-accept rate on spoof trials. Recommended protocols and details are provided in Appendix E.

**ASVSpoof benchmarks.** We evaluate on ASVSpoof-2021 LA and DF (Yamagishi et al., 2021) and ASVSpoof-5 (Wang et al., 2024b) under the strict real-only protocol, reporting AUC (↑) and EER (↓) (%). To reduce overfitting caused by

*Table 1.* AUC (↑) / EER (↓) (%) on ASVSpoof-2021 (LA/DF; DF is further broken down by different synthesizer categories) and ASVSpoof-5 (Whole/VC/TTS/AT) and Transfer. Only the rows below the horizontal rule follow the strict real-only protocol.

| Method | ASVSpoof 2021 LA | ASVSpoof-2021 DF | | | | | | ASVSpoof-5 Test | | | | ASVSpoof-5 Transfer |
|---|---|---|---|---|---|---|---|---|---|---|---|---|
| | | Whole | AR | NAR | TRD | UNK | CONC | Whole | VC | TTS | AT | |
| LCNN | 98.2/ 6.5 | 87.9/21.2 | 86.9/22.3 | 86.0/22.9 | 90.5/18.8 | 88.6/20.5 | 86.8/21.1 | 68.0/36.8 | 80.8/26.9 | 71.1/34.0 | 55.0/46.6 | 65.0/38.9 |
| RawNet2 | 99.1/ 4.2 | 84.4/22.8 | 82.2/25.7 | 83.3/24.2 | 87.5/18.3 | 84.2/22.5 | 84.9/23.8 | 67.3/37.1 | 68.2/36.0 | 75.2/29.4 | 57.6/44.2 | 62.3/40.5 |
| RawGAT | 99.1/ 4.1 | 93.2/14.3 | 88.2/19.5 | 94.1/12.9 | 97.8/ 6.9 | 92.4/15.7 | 94.6/11.8 | 68.7/36.3 | 74.5/32.7 | 71.1/33.7 | 62.4/41.0 | 66.5/39.5 |
| AASIST | 98.5/ 6.2 | 91.8/16.4 | 85.9/21.6 | 91.3/16.4 | 97.9/ 7.0 | 92.1/16.9 | 91.4/16.4 | 72.5/33.7 | 75.7/31.1 | 81.0/24.9 | 63.0/41.1 | 67.5/39.9 |
| MPE | 92.1/16.1 | 81.5/26.2 | 80.4/27.2 | 81.6/26.0 | 84.1/23.9 | 78.5/28.5 | 84.3/24.0 | 73.7/32.1 | 82.1/25.2 | 71.9/32.2 | 70.6/35.2 | 53.9/47.1 |
| ABCNet | 91.8/15.6 | 87.4/21.4 | 81.6/26.3 | 86.4/22.0 | 92.9/15.6 | 88.8/20.4 | 87.2/21.0 | 53.1/49.2 | 48.8/52.3 | 58.4/45.3 | 49.5/51.6 | 63.6/40.7 |
| ASDG | 97.9/ 6.1 | 83.2/25.8 | 81.0/28.1 | 82.9/24.7 | 86.5/22.1 | 82.5/26.4 | 88.9/19.5 | 73.6/32.9 | 79.8/28.2 | 72.5/34.1 | 71.0/34.2 | 56.6/44.9 |
| IG-SVD | **99.6/ 3.0** | 88.9/19.6 | 83.1/24.6 | 88.3/20.1 | 96.5/ 8.5 | 89.7/19.3 | 84.4/22.3 | 66.7/37.5 | 60.8/41.7 | 68.4/35.3 | 68.3/37.1 | 59.8/46.5 |
| OC-SVM | 80.7/26.9 | 71.1/34.8 | 68.2/37.2 | 72.1/34.6 | 74.6/32.7 | 75.8/31.3 | 71.0/34.9 | 61.1/42.4 | 57.2/45.1 | 57.6/45.2 | 69.1/36.5 | 59.8/46.5 |
| Deep-SVDD | 96.5/ 8.8 | 82.5/24.9 | 78.6/28.3 | 83.5/23.7 | 85.9/21.2 | 75.4/30.6 | 80.9/24.9 | 75.4/31.6 | 67.0/38.0 | 69.1/36.6 | 87.6/20.3 | 35.5/60.4 |
| OC-Softmax | 94.9/ 9.0 | 70.0/37.8 | 70.0/37.6 | 69.3/38.5 | 70.9/37.4 | 69.2/38.1 | 71.7/35.2 | 55.1/46.0 | 54.8/46.5 | 54.9/45.9 | 54.9/46.0 | 62.3/43.8 |
| ACS | 92.3/14.7 | 79.3/28.0 | 78.7/28.4 | 79.4/27.7 | 80.5/26.6 | 79.4/27.3 | 73.6/32.8 | 66.0/39.6 | 60.3/42.9 | 66.4/39.3 | 69.8/35.1 | 59.5/44.4 |
| **Ours** | 96.9/ 7.3 | **96.9/ 8.4** | **94.9/11.4** | **97.5/ 6.1** | **97.3/ 6.4** | **95.9/ 9.6** | **98.5/ 3.5** | **92.7/13.4** | **92.7/14.6** | **90.0/17.3** | **96.9/ 8.6** | **88.9/14.7** |

limited bona fide data in individual ASVSpoof-2021 partitions, we merge the bona fide subsets of LA and DF to train a single model and evaluate it on both test sets. For ASVSpoof-5, we train on the official training split. For ASVSpoof-2021, we report LA as a whole and report DF with a breakdown by unseen synthesizer categories: autoregressive (AR), non-autoregressive (NAR), traditional (TRD), unknown (UNK), and concatenation (CONC). For ASVSpoof-5, we additionally report subset results by spoof type: VC, TTS, and adversarial attack (AT).

**Singing voice benchmark.** We further evaluate CA-SOADD on CtrSVDD (Zang et al., 2024), a controlled singing voice deepfake detection benchmark. Different from the ASVSpoof speech benchmarks, CtrSVDD focuses on detecting synthetic singing vocals generated by singing voice synthesis and singing voice conversion systems. We follow its official train, development, and evaluation partitions under the same strict real-only protocol. Since CtrSVDD is used as a standalone singing voice benchmark in our experiments, we train a single-centroid model and report the overall evaluation-set result without further subdividing the test set by attack type or language.

**Cross-lingual benchmark.** We evaluate cross-lingual robustness on MLAAD (Müller et al., 2024) with eight languages: en, de, es, fr, it, pl, ru, and uk. We train a single multilingual model on bona fide utterances pooled across all languages. When the language label is available at test time, we use language-conditioned centroids for scoring.

### 4.2. Implementation Details

We train all models with Adam using a learning rate of $10^{-6}$ and weight decay of $10^{-4}$. Audio preprocessing, sampling, and training schedules are fixed across all experiments. All experiments are run on a single NVIDIA RTX 5090 GPU.

For ASVSpoof benchmarks, we use the official validation split and retain only bona fide utterances. For MLAAD, we create train, validation, and test splits with an 8:1:1 ratio and retain only bona fide utterances in the validation split.

**Warm-up.** We warm up for the first 5 epochs using only $OBJ_{cpt}$ to stabilize the bona fide centroid. Warm-up is omitted only in settings where $OBJ_{binv}$ is not used.

**Probe operator protocol.** We use a unified probe operator pool $\mathcal{A}$ for all benchmarks, without dataset-specific tuning. A sensitivity study that removes operators from $\mathcal{A}$ is reported in Appendix C.

### 4.3. Comparison Baselines

**Baseline protocol scope.** We report two baseline families with explicitly different supervision assumptions. The **spoof-supervised detectors** (above the horizontal rule in Table 1) follow the conventional binary ADD setting and are trained with bona fide and spoof labels, including LCNN (Lavrentyeva et al., 2019), RawNet2 (Jung et al., 2020), RawGAT (Tak et al., 2021), AASIST (Jung et al., 2022), ABC-CapsNet (Wani et al., 2024), MPE (Wang et al., 2024a), ASDG (Xie et al., 2024), and IG_SVD (Ren et al., 2025). These systems are reported *only* as reference detectors and are *outside* the strict real-only protocol. The **real-only baselines** (below the horizontal rule in Table 1) are evaluated under our strict real-only protocol and are trained on bona fide speech only.

Besides, within the real-only group, we further separate: (i) **intrinsically real-only** one-class methods, including OC-SVM (Schölkopf et al., 2001) and Deep-SVDD (Ruff et al., 2018), whose standard training does not require spoof data; (ii) **relaxed one-class methods evaluated in real-only mode**, including OC-Softmax (Zhang et al., 2021) and ACS (Kim et al., 2024), whose original formulations use spoof or negative samples for boundary shaping, but are *adapted* here to real-only training to ensure protocol comparability. For fairness, all four real-only baselines share the same pretrained self-supervised speech encoder as

CA-SOADD, and only differ in the one-class objective and training recipe built on top of the shared representations.

This separation makes the comparison explicit: we measure how much robustness can be achieved under deployable real-only training, while using spoof-supervised detectors only as a non-strict reference upper bound.

# 5. Experiment Results

## 5.1. Unseen Attacks and Benchmark Shift

**Unseen attacks across synthesizer and pipeline shifts.** Table 1 reports results on ASVSpoof-2021 (LA and DF) and on ASVSpoof-5 (whole, VC, TTS, and AT subsets). These partitions encompass complementary shift types. LA and DF represent distinct attack surfaces within the same benchmark, and DF further contains synthesizer-category shifts. For ASVSpoof-5, analyzing specific spoof pipelines (VC, TTS, AT) reveals subset-specific failure modes often masked by aggregate whole-set performance. Across these scenarios, compactness-driven real-only baselines exhibit notable instability. In contrast, our CA-SOADD demonstrates consistent robustness, indicating that strict OCL requires explicit boundary control beyond simple compactness to maintain performance under diverse unseen attacks.

Table 3 further validates the necessity of our tri-objective design. Removing $OBJ_{cabs}$ leads to a marked performance decline, confirming that compactness alone is insufficient to constrain the acceptance region under unseen and diverse attacks. Similarly, removing $OBJ_{binv}$ also degrades detection performance, suggesting that boundary shaping benefits from a locally stable centroid neighborhood; without it, near-centroid regions are more vulnerable to spoof intrusion.

**Benchmark shift via ASVSpoof-2021→ASVSpoof-5 transfer.** To assess zero-shot robustness, we evaluate models trained on ASVSpoof-2021 bona fide speech directly on the ASVSpoof-5 test set under the strict protocol (Table 1).

In this challenging setting, CA-SOADD substantially outperforms all baselines. Notably, it surpasses even spoof-supervised detectors, which suffer severe degradation due to overfitting artifacts specific to the source benchmark. This gap indicates that CA-SOADD avoids memorizing spoof-specific artifacts or benchmark-specific negatives. Instead, its centroid-anchored, margin-based boundary shaping yields a fundamental rejection boundary that generalizes effectively across benchmarks.

**Embedding diagnostics under cosine-to-centroid scoring.** Fig. 3 provides a visual diagnostic of the embedding geometry, comparing ACS and CA-SOADD alongside two ablation models that remove $OBJ_{cabs}$ or $OBJ_{binv}$. On ASVSpoof-5, ACS exhibits significant overlap between bona fide and spoof clusters near the centroid, consistent with its perfor-

mance drop when boundary control is absent. Ablation results show that removing $OBJ_{cabs}$ yields a looser transition region around the centroid, while removing $OBJ_{binv}$ produces a less coherent neighborhood where spoof points more easily drift toward the centroid. In contrast, our CA-SOADD forms a tighter centroid neighborhood with fewer near-centroid spoof points, in line with its better robustness.

## 5.2. Singing Voice Deepfake Detection

**Single-centroid evaluation on CtrSVDD.** We further evaluate CA-SOADD on CtrSVDD, a controlled singing voice deepfake detection benchmark. Following the same protocol as the ASVSpoof benchmarks, we train models on the official training split, select checkpoints using the official development split under the strict real-only protocol, and evaluate on the official evaluation split. Since CtrSVDD is used as a standalone benchmark in our experiments, we report the overall evaluation-set EER without further decomposing the test set by attack type or other subsets.

**Results.** The CtrSVDD column in Table 2 reports the corresponding results. CA-SOADD achieves the best EER of 16.83%, outperforming the strongest competing baseline, AASIST, which obtains 21.81%. This result shows that the proposed centroid-anchored boundary shaping remains effective for singing voice deepfake detection under the same single-centroid real-only evaluation protocol. Compared with compactness-driven real-only baselines such as OC-SVM, Deep-SVDD, OC-Softmax, and ACS, CA-SOADD yields a substantially lower EER, suggesting that explicit boundary shaping is also beneficial when the bona fide distribution consists of singing vocals rather than ordinary speech utterances.

## 5.3. Boundary Shaping is not Surrogate Discrimination

A natural question is whether the gains mainly come from treating distribution-shifted views as explicit negatives and learning a discriminative boundary. To investigate this, we replace our centroid-referenced margin in $OBJ_{cabs}$ with two surrogate objectives—a BCE-based binary classification loss (BCE) and a CSI-style contrastive loss (InfoNCE)—while keeping the identical shift family, strict real-only protocol, and the same cosine-to-centroid inference rule.

Table 3 (middle block) shows that neither surrogate discrimination reproduces the gains: BCE performs consistently worse than the centroid-referenced margin, and InfoNCE is unstable (better than BCE on ASVSpoof-5 but worse on ASVSpoof 2021 LA/DF). This indicates that simply strengthening discriminative/contrastive signals against synthetic shifts does not reliably yield a stable decision boundary under strict OCL. In contrast, the margin-based $OBJ_{cabs}$ ensures robustness by shaping the acceptance region via centroid-referenced constraints, thereby strictly aligning the

*Table 2.* Evaluation results on the MLAAD and CtrSVDD benchmarks. Results are reported as EER (↓) (%). For MLAAD, we report the full set (FULL) and each language subset; for CtrSVDD, we report the corresponding benchmark result.

| METHOD | MLAAD TEST | | | | | | | | | CTRSVDD |
|---|---|---|---|---|---|---|---|---|---|---|
| | FULL | DE | EN | ES | FR | IT | PL | RU | UK | |
| LCNN | **9.42** | **9.95** | **7.88** | **12.31** | **6.12** | 9.57 | 2.50 | 19.08 | **5.94** | 21.87 |
| RAWNET2 | 22.71 | 24.94 | 15.71 | 26.48 | 21.08 | 24.00 | 13.16 | **12.83** | 26.40 | 42.46 |
| RAWGAT | 14.33 | 17.47 | 12.03 | 17.49 | 13.69 | 6.26 | **1.53** | 15.42 | 25.08 | 27.13 |
| AASIST | 14.88 | 16.03 | 11.31 | 20.30 | 14.80 | **5.90** | 1.81 | 18.60 | 27.43 | 21.81 |
| MPE | 12.47 | 14.59 | 9.85 | 16.97 | 14.32 | 9.11 | 8.08 | 20.94 | 8.50 | 23.54 |
| ABCNET | 40.54 | 37.95 | 26.84 | 44.83 | 35.94 | 36.91 | 35.45 | 41.43 | 43.20 | 27.58 |
| ASDG | 10.73 | 14.84 | 8.15 | 16.20 | 8.37 | 7.24 | 3.53 | 28.75 | 7.23 | 27.89 |
| IG-SVD | 19.87 | 22.07 | 13.10 | 26.44 | 24.98 | 18.65 | 8.35 | 18.10 | 21.53 | 29.54 |
| OC-SVM | 43.21 | 45.25 | 47.00 | 50.18 | 46.61 | 38.00 | 32.35 | 48.90 | 36.18 | 45.80 |
| DEEP-SVDD | 26.99 | 37.00 | 18.19 | 37.12 | 33.63 | 29.64 | 23.84 | 32.15 | 42.63 | 39.28 |
| OC-SOFTMAX | 38.07 | 46.16 | 34.09 | 41.69 | 39.78 | 41.07 | 37.22 | 45.75 | 42.68 | 48.69 |
| ACS | 35.94 | 42.52 | 33.34 | 38.92 | 38.22 | 38.43 | 31.05 | 38.55 | 42.71 | 45.52 |
| **OURS (W/O OBJ_{BINV})** | 17.70 | 24.45 | 12.07 | 29.20 | 24.75 | 17.80 | 5.14 | 19.94 | 15.07 | **16.83** |

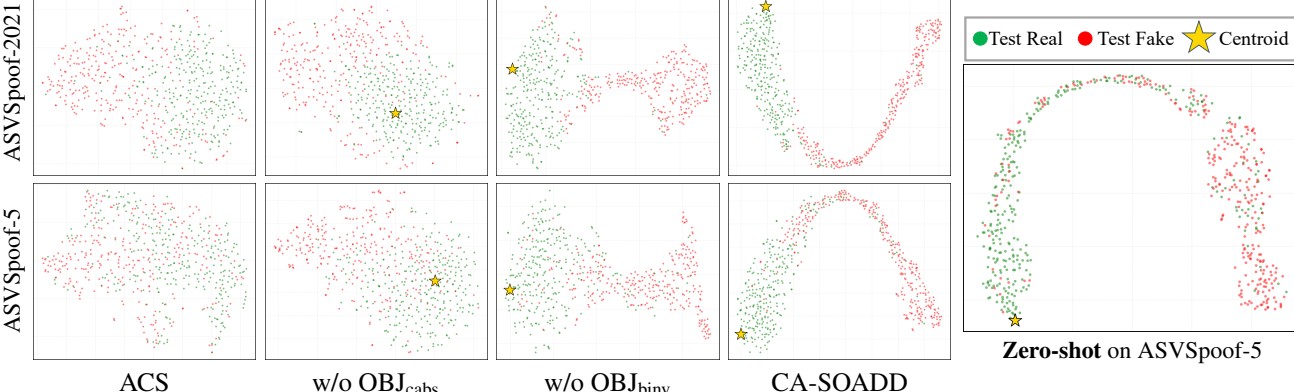

*Figure 3.* t-SNE visualization under the strict real-only protocol for ACS, CA-SOADD, and ablations. The star denotes the training-domain bona fide centroid. The rightmost column corresponds to the zero-shot setting.

training objective with the cosine-to-centroid inference rule. This evidence validates our distinction from CSI (Tack et al., 2020) and NAD (Sinha et al., 2021): our gains arise from score–loss alignment via centroid-anchored boundary shaping, not from learning a classifier against pseudo-negatives.

### 5.4. Handling Heterogeneity via Domain Conditioning

**Language-Specific Centroids.** We evaluate on MLAAD using a single multilingual model trained on bona fide speech from eight languages. Given the multimodal nature of cross-lingual speech, we employ one language-conditioned centroid per language while sharing a unified representation extractor. At test time, trials are scored by cosine similarity to their corresponding language centroid. We assume the test-time language label is available from a standard language identification front-end, so we isolate strict real-only detection under language-induced shifts.

**Default setting on MLAAD.** Under multilingual training, enabling OBJ_{binv} can reduce separation among language-conditioned centroids and hurt performance. We therefore use **CA-SOADD (w/o OBJ_{binv})** as the default on MLAAD, and defer a stronger integration of benign invariance with

*Table 3.* ASVSpoof ablations under the strict real-only protocol. OS=Online Standardization, LS=Layer Selection, TW=Time Weighting. Results are reported as AUC (↑) / EER (↓) (%).

| SETTING | 2021 LA | 2021 DF | ASVSPOOF-5 |
|---|---|---|---|
| W/O OBJ_{CABS} | 96.16 / 9.51 | 89.90 / 18.69 | 72.67 / 33.83 |
| W/O OBJ_{BINV} | 96.80 / 7.99 | 95.24 / 11.56 | 89.12 / 16.43 |
| BCE | 93.88 / 12.58 | 90.76 / 18.39 | 86.33 / 21.56 |
| INFONCE | 91.62 / 15.97 | 85.83 / 24.12 | 89.20 / 16.95 |
| WAVLM | 95.62 / 10.51 | 92.67 / 14.55 | 88.18 / 16.38 |
| HUBERT | 94.98 / 11.08 | 91.31 / 15.35 | 84.73 / 19.90 |
| AUGONLY | 96.36 / 8.59 | 93.84 / 13.32 | 86.40 / 19.15 |
| W/O OS | 94.27 / 12.12 | 93.85 / 14.20 | 89.63 / 17.84 |
| W/O LS | 97.41 / 7.03 | 96.11 / 11.01 | 90.03 / 16.74 |
| W/O TW | 96.08 / 8.59 | 96.23 / 10.48 | 91.69 / 14.88 |
| **OURS** | **96.92 / 7.30** | **96.90 / 8.36** | **92.74 / 13.44** |

cross-language separation to future work.

**Results.** The MLAAD columns in Table 2 present aggregate and per-language results. Overall, language-conditioned centroids yield stable cross-lingual performance on this multi-language anti-spoofing benchmark, validating the necessity of multi-centroid scoring for heterogeneous manifolds where a single centroid risks miscalibration.

## 5.5. Ablations: Ruling Out Alternative Explanations

Table 3 reports ablations under the strict real-only protocol. We structure the analysis as a process of elimination, establishing a four-part exclusion analysis that isolates what drives the gains and what stabilizes centroid-based scoring.

**Gains are not from benign augmentation alone.** The "AugOnly" baseline applies the same benign transform family as CA-SOADD strictly as input-level augmentation, disregarding the benign-view invariance objective. AugOnly consistently underperforms CA-SOADD, exhibiting more pronounced degradations on DF and ASVSpoof-5. This indicates that, in strict real-only detection, robustness requires aligning clean and benign views to stabilize the centroid neighborhood, not merely increasing bona fide diversity.

**Gains are not from surrogate discrimination on shifted views.** As discussed in Sec. 5.3, we explicitly compared the centroid-referenced margin against surrogate discrimination objectives (BCE and CSI-style InfoNCE) under the same shift family and strict real-only protocol. Results show that these variants fail to reproduce the gains, validating that our improvements stem from score–loss alignment via centroid-referenced constraints rather than learning a classifier against shifted views.

**Gains are not tied to a specific SSL backbone.** To verify that the proposed objective is not dependent on a particular self-supervised speech encoder, we replace the default XLSR backbone with WavLM and HuBERT while keeping all other components unchanged. As shown in the third block of Table 3, CA-SOADD remains effective with both alternative backbones. Although XLSR achieves the strongest overall performance, WavLM and HuBERT still yield competitive results under the same strict real-only protocol. This indicates that the observed gains are not merely inherited from a specific pretrained encoder, but are primarily driven by the proposed centroid-anchored loss design and boundary-shaping mechanism.

**Stable centroid scoring requires geometry stabilization.** Finally, the bottom block of Table 3 shows that removing Online Standardization degrades performance, highlighting the importance of real-only statistical calibration for reliable centroid statistics under strict training. Replacing the proposed layer selection or time weighting with uniform pooling yields inferior results, suggesting that lightweight depth–time selection helps produce compact utterance embeddings and stabilizes centroid-based scoring in the absence of spoof supervision.

## 6. Conclusion

We presented CA-SOADD, a strict real-only one-class framework establishing the deployable boundary problem in audio deepfake detection under a single cosine-to-centroid scoring rule. Crucially, we introduce off-manifold boundary probes: instead of treating shifted views as an explicit negative class, CA-SOADD uses them only through centroid-referenced margin constraints, achieving score–loss alignment without surrogate discrimination. Experimental results demonstrate that CA-SOADD yields stronger robustness under unseen attacks, benchmark shift, and heterogeneous bona fide distributions, while fully adhering to strict real-only training, selection, and calibration.

**Data Availability.** We publicly release our code at https://github.com/120L020310/CA-SOADD.

## Acknowledgements

This work was supported by the National Cyber Security-National Science and Technology Major Project under Grant No. 2026ZD1500500, the Key Program of the National Natural Science Foundation of China under Grant No. 62532016, the National Natural Science Foundation of China under Grant No. 62572150, the Guangdong Basic and Applied Basic Research Foundation under Grant No. 2026B1515020033, the National Social Science Fund of China under Grant No. 23VRC094, and the Major Program of the National Social Science Fund of China under Grant No. 22&ZD147.

## Impact Statement

This paper presents work whose goal is to advance the field of Machine Learning. There are many potential societal consequences of our work, none of which we feel must be specifically highlighted here.

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

## A. Baseline Details

This section summarizes the baselines used in our experiments. We include (i) supervised two-class detectors trained with bona fide+spoof labels, and (ii) one-class/real-only baselines trained on bona fide speech only.

### A.1. Supervised two-class detectors

- **LCNN** (Lavrentyeva et al., 2019): A lightweight CNN-based spoofing detector operating on LFCC features.

- **RawNet2** (Jung et al., 2020): An end-to-end waveform-based detector that learns front-end representations directly from raw audio.

- **RawGAT** (Tak et al., 2021): A raw-audio graph-attention model that captures spectro-temporal relations across sub-bands/segments via GAT and graph pooling.

- **AASIST** (Jung et al., 2022): An attention-based architecture tailored for spoofing detection that models discriminative spectro-temporal cues with specialized aggregation.

- **ABC-CapsNet** (Wani et al., 2024): A spectrogram-based method using a CNN feature extractor followed by cascaded capsule networks to model spoofing artifacts.

- **MPE** (Wang et al., 2024a): A feature-augmented approach that injects multiscale permutation entropy cues into an LCNN-style classifier.

- **ASDG** (Xie et al., 2024): A domain-generalization method that learns spoof-robust representations via single-sided adversarial learning with metric constraints.

- **IG_SVD** (Ren et al., 2025): A disentanglement-based framework that separates domain-agnostic and domain-specific artifacts using multi-task/contrastive regularization.

### A.2. One-class / real-only baselines

All one-class baselines are trained on bona fide data only, following the strict real-only protocol, and operate on the same utterance embeddings as our method.

- **OC-SVM** (Schölkopf et al., 2001): A classical kernel one-class method that learns a maximum-margin decision boundary enclosing bona fide embeddings; test scores are given by the signed distance to the learned boundary.

- **Deep-SVDD** (Ruff et al., 2018): A deep one-class method that learns a representation mapping to minimize the volume of a hypersphere centered at $c$ (optionally with a soft boundary radius). The anomaly score is the distance to the center in the learned embedding space.

- **OC-Softmax** (Zhang et al., 2021): A one-class softmax objective that enforces compact bona fide embeddings via angular-margin constraints under a single-class decision rule.

- **ACS** (Kim et al., 2024): An adaptive-center one-class objective that updates the bona fide center using mini-batch statistics and encourages embeddings to be compact around the adaptive center.

## B. Online Standardization: Algorithm and Ablations

**OS vs. LayerNorm/BatchNorm.** LayerNorm is a per-sample normalization: it centers and rescales each embedding independently and does not introduce a global reference frame across samples/batches. In contrast, Online Standardization (OS) defines a *running bona fide reference frame* by maintaining dataset-level running mean/variance estimated from bona fide embeddings only (no gradient) and freezing them at inference. This explicitly aligns embeddings to a coordinate system anchored on bona fide statistics, which is structurally compatible with centroid-based scoring and centroid updates under the strict real-only protocol.

**Pathology diagnosis: cosine collapse before OS.** We quantify the "cosine-collapse" phenomenon reported in the main text: after the geometry-stabilized extractor and projection head $g(\cdot)$ (before OS), bona fide embeddings become highly

*Table 4.* Diagnosis of cosine-collapse and the effect of OS. "Before OS" denotes embeddings after the projection head $g(\cdot)$; "After OS" denotes embeddings after RunningStandardize. Pairwise cosine is computed over $K{=}100,000$ random bona fide pairs; centroid cosine percentiles are computed over all bona fide samples.

| SETTING | PAIRWISE MEAN | PAIRWISE STD | $q_{50}(\cos(\boldsymbol{z}, \boldsymbol{c}))$ | $q_{90}$ | $q_{99}$ |
|---|---|---|---|---|---|
| BEFORE OS | 0.9977 | 0.0135 | 0.9997 | 0.9998 | 0.9998 |
| AFTER OS | 0.9205 | 0.1330 | 0.9744 | 0.9839 | 0.9903 |

*Table 5.* Ablation on the OS module under the strict real-only protocol. All settings replace the original OS module with an alternative module while keeping the rest of the system unchanged. Results are reported as AUC ($\uparrow$) / EER ($\downarrow$) (%).

| MODULE AT OS POSITION | 2021 LA | 2021 DF | ASVSPOOF-5 |
|---|---|---|---|
| W/O OS | 94.27 / 12.12 | 93.85 / 14.20 | 89.63 / 17.84 |
| EXTRA-LN | 94.44 / 12.31 | 93.92 / 14.08 | 89.59 / 17.95 |
| RMSNORM | 95.88 / 10.40 | 94.79 / 12.41 | 90.29 / 16.90 |
| BN(BF) | 96.02 / 8.68 | 94.60 / 11.81 | 90.67 / 15.82 |
| BN(ALL) | 92.44 / 14.39 | 91.98 / 15.20 | 88.83 / 18.56 |
| OS (OURS) | **96.92 / 7.30** | **96.90 / 8.36** | **92.74 / 13.44** |

concentrated, yielding near-saturated cosine similarities and making centroid-anchored separation difficult. We compute two diagnostics on bona fide training/development embeddings: (i) *pairwise cosine* statistics of $\cos(\tilde{\boldsymbol{z}}_i, \tilde{\boldsymbol{z}}_j)$ over $K$ uniformly sampled pairs, and (ii) *centroid cosine* percentiles of $\cos(\tilde{\boldsymbol{z}}_i, \tilde{\boldsymbol{c}})$ over all bona fide samples, where $\tilde{\boldsymbol{z}} = \boldsymbol{z}/\|\boldsymbol{z}\|_2$ and $\tilde{\boldsymbol{c}} = \boldsymbol{c}/\|\boldsymbol{c}\|_2$ match the cosine scoring rule. Table 4 shows that OS substantially increases angular dispersion (larger std, lower high-percentile saturation), i.e., it "opens up" the geometry while remaining anchored on real-only statistics. Pairwise cosine is computed from $K = 100,000$ random bona fide pairs; the centroid is computed from bona fide training embeddings and reused for dev.

**Placement and protocol.** OS is applied *after* the projection head $g(\cdot)$ (a lightweight MLP that already includes LayerNorm) and thus acts as a bona fide-referenced coordinate alignment before centroid estimation and cosine scoring. All ablations below keep the backbone, layer/time aggregation, and the projection head fixed, and only replace the *module at the OS insertion point*.

**Irreplaceability at the OS insertion point.** To test whether OS is merely "another normalization choice", we perform a same-location replacement study by replacing *only* the original OS module inserted after the projection head $g(\cdot)$, while keeping the backbone, layer/time aggregation, $g(\cdot)$ (including its internal LayerNorm), and all training hyperparameters fixed. "w/o OS" removes OS (identity mapping). "Extra-LN" inserts an additional LayerNorm at the same position; "RMSNorm" inserts RMSNorm. "BN(bf)" inserts BatchNorm whose running statistics are updated using *only* bona fide samples to respect the strict real-only protocol, whereas "BN(all)" uses standard batch composition and is thus not strictly real-only.

As shown in Table 5, per-sample normalizers at this position (Extra-LN and RMSNorm) provide limited gains over removing OS, and BN(bf) improves performance but still falls short of OS. In contrast, OS yields a clear and consistent improvement across all benchmarks, suggesting that the benefit is not simply from adding another normalization layer, but from aligning embeddings to a bona fide-defined *running* reference frame that is structurally matched to centroid-anchored one-class scoring and online centroid estimation. The poor performance of BN(all) further highlights the importance of real-only statistics: mixing non-bona-fide samples in batch statistics can distort the reference frame under strict real-only training.

# C. Distribution-Shifted View Design: Relation to Shifted/Negative Augmentation

This section clarifies the relationship between our distribution-shifted view design and prior shifted/negative augmentation paradigms. While both lines of work construct distributionally shifted views from bona fide speech, our method uses such views *only* as off-manifold boundary probes under a strict real-only protocol. All discussions and comparisons in this section assume the same strict real-only setting and the same centroid-similarity scoring rule as in the main paper.

## C.1. Relationship to shifted-instance and negative augmentation paradigms

Shifted-instance and negative augmentation paradigms introduce distributionally shifted samples constructed from in-distribution data. Our distribution-shifted view construction follows the same high-level idea of creating shifted views, but

---

**Algorithm 1** Online Standardization (OS) with real-only online statistics

---

**Input:** embeddings $X \in \mathbb{R}^{B \times D}$, labels $y \in \{0, 1\}^B$ (1=bona fide), flag $update\_stats$
**Buffers:** $n \in \mathbb{N}$, $\mu \in \mathbb{R}^D$, $M2 \in \mathbb{R}^D$, constants $\epsilon$, $use\_var$
**Output:** standardized embeddings $\hat{X} \in \mathbb{R}^{B \times D}$
**if** $update\_stats$ **then**
    $X_{bf} \leftarrow \{x_i : y_i = 1\}$
    **if** $|X_{bf}| > 0$ **then**
        $b \leftarrow |X_{bf}|$
        $\mu_b \leftarrow \frac{1}{b} \sum_{x \in X_{bf}} x$
        **if** $use\_var$ **then**
            $M2_b \leftarrow \sum_{x \in X_{bf}} (x - \mu_b) \odot (x - \mu_b)$
        **else**
            $M2_b \leftarrow 0$
        **end if**
        $n_0 \leftarrow n$, $\ n_1 \leftarrow n_0 + b$
        **if** $n_0 = 0$ **then**
            $\mu \leftarrow \mu_b$, $\ M2 \leftarrow M2_b$, $\ n \leftarrow n_1$
        **else**
            $\delta \leftarrow \mu_b - \mu$
            $\mu \leftarrow \mu + \delta \cdot \frac{b}{n_1}$
            **if** $use\_var$ **then**
                $M2 \leftarrow M2 + M2_b + (\delta \odot \delta) \cdot \frac{n_0 b}{n_1}$
            **end if**
            $n \leftarrow n_1$
        **end if**
    **end if**
**end if**
$\hat{X} \leftarrow X - \mu$                     (broadcast over batch)
**if** $use\_var$ **and** $n > 1$ **then**
    $\sigma^2 \leftarrow M2/(n-1)$
    $\hat{X} \leftarrow \hat{X} \oslash \sqrt{\sigma^2 + \epsilon}$            (element-wise)
**end if**
**return** $\hat{X}$

---

differs in *how* these views are used during training and *how* the detector is scored at inference.

**Training objective: boundary shaping versus surrogate discrimination.** Our framework does not treat distribution-shifted views as an explicit negative class. Instead, they enter training only through a centroid-referenced margin term as off-manifold probes that tighten the strict one-class acceptance region around bona fide statistics. To verify that the benefit is not from surrogate discrimination, we include an objective-form ablation in Sec. 5.3, where only the objective form is changed while keeping the same bona fide data and the same shifter family $\mathcal{A}$.

**Inference rule: centroid similarity (objective–inference alignment).** All results reported in this work use a single scoring rule: cosine similarity between the test embedding and the (domain-conditioned) bona fide centroid. We do not use alternative non-parametric scoring rules in this paper, ensuring that the training signal is aligned with the inference metric.

## C.2. Structural off-manifold probe synthesis

We define **structural off-manifold probe synthesis** as designing the shifter family $\mathcal{A}$ to generate views that disrupt speech structures relevant to authenticity, without relying on spoof/attack examples. The goal is to provide diverse off-manifold boundary probes through complementary disruption mechanisms:

- **Temporal structure disruption** (e.g., segment permutation),

- **Spectral/bandwidth disruption** (e.g., spectral masking, resampling-induced bandwidth shifting),

- **Phase/coherence disruption** (e.g., phase scrambling),

- **Reconstruction-style perturbations** that mimic analysis–quantization–reconstruction or weak vocoder inconsistencies (e.g., PNCR, VocHF),

- **Identity-invariant perturbation** to diversify probe directions without introducing spoof priors (IDInv).

These operators are not intended to approximate any specific spoof generator. They provide generic off-manifold perturbations that can tighten the one-class acceptance region when coupled with the centroid-anchored margin term.

### C.3. Unified operator pool and implementation details

**Operator pool.** We use a single fixed shifter family $\mathcal{A}$ across all benchmarks (ASVSpoof and MLAAD), without dataset-specific modification. For each bona fide utterance and each forward pass, we sample exactly one operator $a \sim \mathcal{A}$ and apply it to synthesize a shifted view $a(x)$. The operator is sampled *once per mini-batch* (i.e., the same operator is applied to all samples in that forward pass), matching our implementation. In the main text, $\mathcal{A}(\cdot)$ denotes sampling an operator $a$ from a probe family and applying $a(\cdot)$; here we instantiate the family as a finite operator pool $\mathcal{A} = \{a_k\}$.

Concretely,
$$\mathcal{A} = \{\text{PNCR, VOCHF, IDINV, PERM, SPECMASK, PHASESCR, RESAMP}\}. \tag{12}$$

**Operator definitions (implementation-faithful).** (PNCR) *Pseudo neural codec resynthesis*: analyze the waveform using a sinc bandpass filterbank (8 bands, kernel 63, stride 4), apply $\mu$-law companding ($\mu$=255) and uniform quantization (6 bits; straight-through estimator), then reconstruct via transposed-convolution upsampling and low-pass filtering, with a mild periodic boundary envelope. (VOCHF) *Weak vocoder-like HF artifacts*: split low/high bands with a sinc low-pass (kernel 63), apply a weak comb-style HF emphasis (stride 4), sinusoidal micro phase-delay modulation, and short-kernel HF smoothing. (IDINV) *Identity-invariant perturbation*: randomly apply one of {random bandlimiting via a sinc low-pass bank, additive noise at 28–40 dB SNR, soft clipping with gain 1.2–2.5}. (PERM) *Segment permutation*: divide the waveform into 8 equal segments and randomly permute them. (SPECMASK) *Spectral masking*: apply rFFT and zero out one contiguous frequency band with ratio 0.1, then inverse FFT. (PHASESCR) *Phase scrambling*: preserve rFFT magnitudes but replace phases with i.i.d. uniform random phases, then inverse FFT. (RESAMP) *Resample shift*: downsample by a fixed *ratio* using linear interpolation and resample back to the original length, ensuring a non-trivial bandwidth shift across different sampling rates.

### C.4. Controlled probe-generator paradigm study (diagnostic)

We conduct a controlled diagnostic to compare different *probe-generation paradigms* for centroid-referenced boundary shaping. This study is diagnostic (not used for dataset-specific tuning): throughout the paper we use the fixed structural probe pool $\mathcal{A}$ (Sec. C.3) for $\text{OBJ}_{\text{cabs}}$. Here we replace the probe generator with alternative families while keeping the strict real-only protocol, backbone, real-only data, optimizer, schedule, and centroid-similarity scoring fixed.

**Isolating the boundary-shaping signal.** In the main protocol, $\text{OBJ}_{\text{binv}}$ enforces invariance by pulling representations of $x$ and $t(x)$ (with $t \sim \mathcal{T}$) closer. If we simultaneously treat the same $\mathcal{T}$ as boundary probes, the objectives can impose conflicting gradients (pulling $t(x)$ toward the centroid while also pushing it away by a margin), making the comparison uninterpretable. We therefore disable $\text{OBJ}_{\text{binv}}$ in this diagnostic so that training depends only on $\text{OBJ}_{\text{cpt}}+\text{OBJ}_{\text{cabs}}$ and the chosen probe family. This diagnostic is *not* used to select the main setting; the main experiments follow the protocol described in the paper ($\text{OBJ}_{\text{binv}}$ enabled for ASVSpoof and disabled for MLAAD as stated in Sec. 5.4).

**F1: RobustAug ($\mathcal{R}$, robustness augmentations).** We use a broader robustness augmentation family $\mathcal{R}$ as probes, including additive noise (with random SNR), mild reverberation (short exponential impulse response), speed perturbation (resample + crop/pad), random low-pass filtering, and soft clipping. This family targets channel/codec/recording variability and is intentionally broader than the invariance transforms used by $\text{OBJ}_{\text{binv}}$.

**F2: BenignAug ($\mathcal{T}$, invariance transforms; implementation-faithful).** We use the same benign transformation family $\mathcal{T}$ as in $\text{OBJ}_{\text{binv}}$. Concretely, $\mathcal{T}$ applies (i) random amplitude scaling (scale $\in [0.5, 1.5]$), (ii) additive white Gaussian noise with fixed relative intensity (noise rms = $0.05\times$ signal rms, i.e., $\approx$26 dB SNR), and (iii) a contiguous time-domain cutout masking 10% of the samples. Note that $\mathcal{T}$ may overlap with $\mathcal{R}$ in operator type (e.g., additive noise), but differs in scope and parameterization.

*Table 6.* Distribution-shifted generator ablation under the strict one-class objective (real-only). Only the probe generator is changed; all other components and centroid-similarity scoring are fixed. Results are AUC (↑) / EER (↓) (%).

| PROBE-GENERATOR | 2021 LA | 2021 DF | ASVSPOOF-5 | MLAAD |
|---|---|---|---|---|
| ROBUSTAUG (F1) | 75.08 / 27.52 | 78.35 / 24.74 | 67.90 / 33.41 | 60.08 / 44.86 |
| BENIGNAUG (F2) | 94.58 / 11.72 | 87.57 / 22.21 | 80.73 / 26.90 | 77.89 / 29.57 |
| STRUCTPROBE (F3) | **96.92 / 7.30** | **96.90 / 8.36** | **92.74 / 13.44** | **86.49 / 17.70** |

**F3: StructProbe ($\mathcal{A}$, structural off-manifold probes).** We use the fixed structural probe pool $\mathcal{A}$ defined in Sec. C.3. $\mathcal{A}$ is specified independently of spoof data and is kept unchanged across datasets. It contains complementary structure-disruption mechanisms (temporal, spectral/bandwidth, phase/coherence, reconstruction-style, and IDINV) that generate distribution-shifted views used *only* as off-manifold boundary probes in the centroid-referenced margin term.

Table 6 reports the results. RobustAug ($\mathcal{R}$) provides weak boundary signals under centroid-anchored strict OCL, and BenignAug ($\mathcal{T}$) yields only partial gains, suggesting that generic robustness or invariance transforms are insufficient to reliably tighten the acceptance region. In contrast, StructProbe ($\mathcal{A}$) performs consistently better, supporting that authenticity-relevant structural disruptions provide stronger off-manifold probe signals for centroid-referenced margin shaping.

## C.5. Probe sensitivity by disruption mechanism

A remaining concern is whether performance gains hinge on a particular operator that accidentally resembles spoof artifacts. We therefore evaluate sensitivity by removing one disruption-mechanism group at a time from $\mathcal{A}$, including temporal, spectral/bandwidth, phase/coherence, reconstruction-style, and IDINV probes, while keeping the strict real-only protocol, objectives, optimization, and centroid-similarity inference fixed. As shown in Table 7, removing any single mechanism causes moderate but not catastrophic changes, indicating that the gains do not rely on one specific "spoof-like" transform. Instead, they arise from the centroid-margin probe mechanism coupled with complementary off-manifold perturbations. We also observe benchmark-dependent preferences: for example, some reduced variants remain competitive on individual benchmarks, while others degrade more clearly under stronger condition shifts. This diagnostic is intended to reveal such sensitivity rather than to motivate dataset-specific probe tuning.

We additionally report representative alternative probe combinations to examine whether the full probe pool is merely exploiting a particular disruption operator. The results show that several reduced combinations remain competitive on individual benchmarks, but none consistently surpasses the full $\mathcal{A}$ across all settings. This behavior is consistent with our design motivation: the probes are not intended to match specific spoofing methods. Instead, they are designed to cover generic manipulation traces commonly observed in fake or degraded audio, including temporal discontinuity, spectral information loss, phase inconsistency, bandwidth limitation, codec-induced reconstruction artifacts, vocoder-related high-frequency anomalies, and channel/compression perturbations. Therefore, the probe pool is intentionally redundant rather than specialized. The reduced-probe results suggest that the learned decision boundary is not tightly coupled to particular known attack types; rather, it is encouraged to capture a broader set of off-manifold directions associated with generic forgery characteristics. For an unseen attack, the generated audio may still exhibit part of these generic characteristics and can therefore be constrained by at least some probes in $\mathcal{A}$. This may help explain why CA-SOADD maintains robustness under attack mismatch, and is also consistent with its strong performance on the newly introduced CtrSVDD benchmark. From a practical perspective, this indicates a weaker deployment assumption: CA-SOADD can be trained and applied without spoof data, while still achieving robust performance against unseen attacks compared with supervised and relaxed one-class baselines.

Overall, the supplementary combination results reinforce the conclusion from the leave-one-mechanism analysis. The full $\mathcal{A}$ provides the most stable trade-off across ASVSpoof-2021 LA, ASVSpoof-2021 DF, and ASVSpoof-5, while individual mechanism removals or reduced combinations lead to benchmark-dependent fluctuations. Therefore, the effectiveness of CA-SOADD is better explained by the joint effect of multiple complementary probes under centroid-anchored boundary shaping, rather than by a single dominant disruption operator or a dataset-specific probe choice.

## C.6. Summary

Our distribution-shifted view design is related to shifted/negative augmentation in that it constructs shifted views from bona fide speech. It differs in training usage: shifted views are used only as off-manifold probes in a centroid-referenced margin

*Table 7.* Probe sensitivity under the strict real-only protocol. $\mathcal{A}$ is the fixed probe pool used throughout the paper. The first block removes a *mechanism group* from $\mathcal{A}$ while keeping all other components unchanged, and the second block reports representative probe combinations. Results are AUC ($\uparrow$) / EER ($\downarrow$) (%).

| PROBE POOL VARIANT | 2021 LA | 2021 DF | ASVSPOOF-5 | MLAAD |
|---|---|---|---|---|
| *Leave-one-mechanism-out variants* | | | | |
| W/O TEMPORAL (PERM) | 96.77 / 7.67 | 95.78 / 8.89 | 91.96 / 14.35 | 86.90 / **17.29** |
| W/O SPECTRAL/BW (SPECMASK+RESAMP) | 95.48 / 8.77 | 94.61 / 10.14 | 91.28 / 15.34 | 86.33 / 17.95 |
| W/O PHASE/COHERENCE (PHASESCR) | 96.31 / 8.19 | 95.89 / 9.40 | 92.08 / 13.96 | 85.78 / 18.01 |
| W/O RECONSTRUCTION (PNCR+VOCHF) | 96.11 / 8.42 | 96.76 / **8.09** | 92.43 / 13.57 | 84.75 / 20.08 |
| W/O IDINV | 96.53 / 7.95 | 96.38 / 8.91 | 92.02 / 13.89 | **86.92** / 17.41 |
| *Alternative probe combinations* | | | | |
| PERM+SPECMASK+PHASESCR+IDINV | 96.24 / 8.06 | 95.88 / 9.03 | 92.25 / 13.72 | 85.98 / 18.21 |
| PHASESCR+RESAMP+PNCR+VOCHF | 95.98 / 8.18 | 96.27 / 8.86 | 92.43 / 13.90 | 85.40 / 18.66 |
| PNCR+RESAMP+VOCHF+IDINV | 95.89 / 8.51 | 95.72 / 9.11 | 91.86 / 13.86 | 86.88 / 17.89 |
| PERM+RESAMP | 96.74 / 7.79 | 96.11 / 9.15 | 92.22 / 13.73 | 84.66 / 20.16 |
| PNCR+VOCHF | 95.90 / 8.85 | 95.65 / 10.08 | 91.05 / 15.42 | 85.03 / 19.32 |
| PERM+IDINV | 95.56 / 8.94 | 95.82 / 9.86 | 91.94 / 14.06 | 85.89 / 18.77 |
| $\mathcal{A}$ (OURS) | **96.92** / **7.30** | **96.90** / 8.36 | **92.74** / **13.44** | 86.49 / 17.70 |

term to shape a strict one-class rejection boundary, rather than to train a surrogate discriminative objective. All experiments adopt a unified shifter family $\mathcal{A}$ (a fixed operator pool sampled once per mini-batch) across benchmarks, and sensitivity diagnostics quantify robustness to probe design without relying on spoof priors.

## D. Spoof-family Shift Robustness (LOFO on ASVSpoof-2021 DF)

A key concern for spoof-supervised detectors is spoof-family dependence: a model trained with access to spoof examples may overfit to generator-/family-specific artifacts, leading to a substantial performance drop when encountering unseen spoof families. To probe this effect in a controlled manner, we conduct a spoof-family shift evaluation on ASVSpoof-2021 DF using a **leave-one-family-out (LOFO)** style protocol, where "family" is proxied by the DF subset categories {AR, NAR, TRD, CONC}.[1]

**Protocol.** For each held-out family $f \in \{\text{AR}, \text{NAR}, \text{TRD}, \text{CONC}\}$, we remove all spoof trials belonging to $f$ from the ASVSpoof-2021 DF training set, and train the spoof-supervised baselines on the remaining spoof families (together with bona fide speech). We then evaluate the trained model on the test subset of the held-out family $f$. This isolates the effect of *family shift* while keeping the dataset, evaluation pipeline, and acoustic conditions fixed. Importantly, our method follows the strict real-only protocol and never uses spoof data for training; therefore, our training procedure is identical across all LOFO splits, and we report the same performance on each subset for our method.

**Results.** Table 8 reports EER on each spoof-family subset. Spoof-supervised baselines exhibit consistent degradation under family shift: when a family is excluded from training, EER on the corresponding held-out subset increases noticeably, with particularly severe drops on TRD for several methods (e.g., RawGAT and AASIST), indicating strong family-dependent decision boundaries. In contrast, our strict real-only detector remains stable across subsets and achieves the lowest EER on all four categories. These results support our hypothesis that learning a centroid-anchored strict one-class acceptance region from bona fide speech mitigates reliance on family-specific spoof artifacts, improving robustness under spoof-family shift.

## E. Bonafide-only Threshold Calibration

AUC and EER are widely used in ASVSpoof-style benchmarks, but EER corresponds to a *two-class* threshold that depends on both bona fide and spoof scores. Under the strict real-only protocol, spoof data are unavailable for calibration, so we provide a real-only procedure to select a decision threshold without using spoof labels.

**Definitions (FAR/FPR and FRR/FNR).** Let $s(x) = \text{sim}(z, c)$ denote the cosine-to-centroid similarity, where larger scores indicate higher confidence of being bona fide. Given a threshold $\tau$, we accept $x$ as bona fide iff $s(x) \geq \tau$. We evaluate two error rates: (i) **FRR** (also called **FNR**): the fraction of *bona fide* trials rejected, i.e., $\mathbb{P}[s(x) < \tau \mid y = \text{bona fide}]$; (ii) **FAR** (also called **FPR**): the fraction of *spoof* trials accepted, i.e., $\mathbb{P}[s(x) \geq \tau \mid y = \text{spoof}]$. In the table, we report **FPR/FAR** and

---

[1] The UNK category is not included because it does not appear as a training subset in our setup.

*Table 8.* LOFO results EER(↓) (%) on on ASVSpoof-2021 DF. Values in parentheses indicate the performance gap compared to the standard test set.

| Model | ASVSpoof-2021 DF Subset (EER %) | | | |
|---|---|---|---|---|
| | AR | NAR | TRD | CONC |
| LCNN | 24.46(↑2.18) | 24.64(↑1.74) | 26.39(↑ 7.58) | 22.54(↑1.49) |
| RawNet2 | 29.59(↑3.89) | 26.03(↑1.85) | 27.66(↑ 9.32) | 25.13(↑1.38) |
| RawGAT | 22.91(↑3.44) | 16.76(↑3.87) | 20.51(↑13.57) | 19.98(↑8.17) |
| AASIST | 22.90(↑1.32) | 18.83(↑2.45) | 29.69(↑22.74) | 21.49(↑5.14) |
| MPE | 32.21(↑5.00) | 26.85(↑0.85) | 28.93(↑ 5.04) | 27.55(↑3.56) |
| ABCNet | 23.81(↓2.49) | 24.30(↑2.31) | 34.97(↑19.41) | 22.72(↑1.72) |
| ASDG | 31.18(↑3.09) | 27.31(↑2.62) | 31.20(↑ 9.08) | 28.38(↑8.85) |
| IG-SVD | 28.20(↑3.60) | 25.07(↑4.93) | 26.42(↑17.92) | 25.74(↑3.40) |
| **CA-SOADD** | **11.43(↑0.00)** | **6.05(↑0.00)** | **6.38(↑ 0.00)** | **3.45(↑0.00)** |

**FNR/FRR** in percent.

**Bonafide-only threshold rule.** We use a real-only development split (from the same training pipeline) to obtain bona fide scores $\{s_i\}$. For a target tail level $\alpha$, we set the threshold as the bottom-$\alpha$ quantile of bona fide scores:

$$\tau_\alpha = Q_\alpha\big(s(\text{bona fide-dev})\big). \tag{13}$$

Intuitively, $\tau_\alpha$ discards the lowest-scoring $\alpha$ fraction of bona fide samples on the development split. By construction, the empirical development FRR approximately equals $\alpha$ (up to finite-sample discretization).

**Key point: $\alpha$ is a *calibration knob*, not a test-set guarantee.** A common confusion is to expect $\alpha$ to equal the achieved test-set FRR/FNR. This is *not* guaranteed: $\alpha$ controls the tail on the *bona fide development split*, while the achieved test-set FNR depends on how bona fide scores shift between development and test (channel, speaker, content, codec, and other domain factors). Therefore, we always report the realized (FPR, FNR) on the test set at $\tau_\alpha$.

**Results and interpretation.** Table 9 reports FPR/FAR and FNR/FRR on ASVSpoof-2021 LA/DF and ASVSpoof-5 using $\alpha \in \{10\%, 5\%, 1\%, 0.5\%, 0.1\%\}$. As expected, decreasing $\alpha$ lowers the threshold (accepting more trials), which reduces FNR but increases FPR. The cross-benchmark differences are substantial: for the same $\alpha$, ASVSpoof-5 yields markedly higher FPR than ASVSpoof-2021, indicating that ASVSpoof-5 spoofs are closer to the bona fide centroid under the same scoring rule and thus require a stricter operating point to control false accepts.

**How to choose $\alpha$ for an unseen target domain.** Since spoof data are unavailable for threshold tuning, $\alpha$ should be treated as an *application-level operating point*: a smaller $\alpha$ prioritizes user experience (lower FRR/FNR) at the expense of security (higher FAR/FPR), while a larger $\alpha$ prioritizes security at the expense of user experience. For a previously unseen target domain, the recommended practice is to: (1) report a sweep over multiple $\alpha$ values (as in Table 9) to expose the FRR–FAR trade-off without spoof calibration; and (2) if possible, collect a small real-only calibration split from the target domain and re-estimate $\tau_\alpha$ *in-domain*. This keeps the protocol real-only while substantially reducing threshold mismatch caused by domain shift.

**Why ASVSpoof-5 and ASVSpoof-2021 require different thresholds.** The table shows that a single fixed $\alpha$ does not yield the same FAR/FPR across benchmarks. This is expected under benchmark shift: the score distributions of both bona fide and spoof can move, changing how conservative a given $\tau_\alpha$ is. We therefore recommend treating $\tau_\alpha$ as *domain-conditioned* whenever in-domain bona fide calibration data are available, and otherwise reporting multiple $\alpha$ levels as standardized "risk tiers".

## F. Hyperparameter Sensitivity

### F.1. Sensitivity to Loss Weights

This section evaluates the sensitivity of CA-SOADD to the loss-weight configuration. We perturb the relative weights while keeping the extractor, training protocol, and all other hyperparameters fixed. All experiments are conducted under the same strict real-only setting on ASVSpoof benchmarks.

*Table 9.* Bonafide-only threshold calibration under the strict real-only protocol. The decision threshold $\tau_\alpha$ is set to the $\alpha$-quantile of similarity scores on a real-only development split, i.e., the bottom-$\alpha$ tail. We report FPR ($\downarrow$) / FNR($\downarrow$) (%) on each test set at the resulting threshold.

| $\alpha$ (BONAFIDE TAIL) | 2021 LA | 2021 DF | ASVSPOOF-5 |
|---|---|---|---|
| 0.1% | 31.21 / 2.71 | 19.35 / 3.36 | 67.41 / 1.24 |
| 0.5% | 13.96 / 4.69 | 9.16 / 8.10 | 47.02 / 3.13 |
| 1% | 9.35 / 5.81 | 6.46 / 12.07 | 39.01 / 4.20 |
| 5% | 3.40 / 12.29 | 2.13 / 29.36 | 19.70 / 8.80 |
| 10% | 2.30 / 18.79 | 1.16 / 38.69 | 14.02 / 13.45 |

*Table 10.* Sensitivity of Recipe-C to loss weights on ASVSpoof benchmarks (real-only). Results are AUC ($\uparrow$) / EER($\downarrow$) (%). The default setting is $(\lambda_1, \lambda_2, \lambda_3) = (5, 1, 1)$.

| $(\lambda_1, \lambda_2, \lambda_3)$ | 2021 LA | 2021 DF | ASVSPOOF-5 |
|---|---|---|---|
| $(1, 1, 1)$ | 96.76 / 7.64 | 94.38 / 12.80 | 85.58 / 18.75 |
| $(2, 1, 1)$ | 96.81 / 7.48 | 96.02 / 10.18 | 92.44 / 13.58 |
| $(5, 1, 1)$ | **96.92 / 7.30** | **96.90 / 8.36** | **92.74 / 13.44** |
| $(10, 1, 1)$ | 95.80 / 9.06 | 96.13 / 10.20 | 89.74 / 16.63 |
| $(5, 2, 2)$ | 96.06 / 8.86 | 94.78 / 11.31 | 92.08 / 14.33 |
| $(5, 1, 2)$ | 96.61 / 7.96 | 96.44 / 8.83 | 92.38 / 14.22 |
| $(5, 0.5, 2)$ | 96.32 / 8.77 | 96.18 / 9.39 | 92.92 / 13.65 |
| $(5, 1, 5)$ | 96.48 / 8.34 | 95.81 / 10.39 | 92.47 / 14.31 |
| $(3, 1, 3)$ | 93.82 / 11.19 | 96.20 / 9.69 | 92.34 / 14.13 |
| $(2, 1, 2)$ | 96.59 / 7.98 | 93.29 / 12.00 | 91.06 / 14.41 |

**Setup.** Recall that CA-SOADD optimizes a weighted combination of the three terms:

$$\mathcal{L} = \lambda_1 \mathcal{L}_{cpt} + \lambda_2 \mathcal{L}_{binv} + \lambda_3 \mathcal{L}_{cabs} + \lambda_4 \mathcal{L}_{ent}. \tag{14}$$

Unless otherwise stated, we report AUC ($\uparrow$) / EER ($\downarrow$) (%) on LA, DF, and ASVSpoof-5. For a fair comparison, each setting uses identical training schedules and evaluation protocols; only $(\lambda_1, \lambda_2, \lambda_3)$ changes.

**Weight perturbations and discussion.** We assess sensitivity in two complementary ways: (i) varying the relative importance of the centroid-compactness term $\mathcal{L}_{cpt}$ while keeping $(\lambda_2, \lambda_3)$ fixed, and (ii) perturbing the weights of the invariance term $\mathcal{L}_{binv}$ and the boundary-shaping term $\mathcal{L}_{cabs}$ around the default configuration, including settings that emphasize $\mathcal{L}_{cabs}$ and settings that rescale multiple terms simultaneously. Table 10 shows that CA-SOADD maintains a reasonably wide stable region around the default weights: moderate perturbations lead to only minor performance fluctuations across LA, DF, and ASVSpoof-5. In contrast, overly weakening or overly amplifying $\mathcal{L}_{cpt}$ (e.g., $(1, 1, 1)$ or $(10, 1, 1)$), or excessively up-weighting $\mathcal{L}_{cabs}$ together with other reweighting changes, can degrade performance on some benchmarks. Overall, these results indicate that CA-SOADD is not dependent on a narrowly tuned weight choice, and they suggest a robust regime where $\mathcal{L}_{cpt}$ remains the primary term while $\mathcal{L}_{binv}$ and $\mathcal{L}_{cabs}$ provide complementary regularization.

### F.2. Margin sensitivity

The margin $m$ in OBJ$_{cabs}$ controls the required cosine-similarity gap between a bona fide embedding and its off-manifold probe relative to the centroid. It therefore determines the strength of boundary tightening under the same cosine-to-centroid geometry used at inference. A small margin provides only a weak boundary-shaping signal, while an overly large margin may impose an excessively restrictive constraint and make optimization harder. We set $m = 1.0$ as a moderate default in all main experiments.

To examine the sensitivity to this hyperparameter, we conduct a margin ablation on ASVSpoof-2021 under the strict real-only protocol. As shown in Table 11, performance is relatively stable around the default setting. For $m \in [0.5, 1.9]$, both LA and DF maintain competitive performance, with the best overall result obtained at $m = 1.0$. When $m$ is too small, e.g., $m = 0.1$, the boundary-shaping signal becomes weaker, leading to degraded performance especially on DF. When $m$ becomes too large, e.g., $m = 1.95$, performance drops on both LA and DF, suggesting that over-tightening the acceptance boundary can harm optimization and reduce generalization. Overall, CA-SOADD is moderately sensitive to the margin but not fragile around the default choice.

*Table 11.* Margin ablation under the strict real-only protocol. Results are reported as AUC (↑) / EER (↓) (%).

| MARGIN | 2021 LA | 2021 DF |
|---|---|---|
| 0.1 | 96.69 / 7.44 | 94.85 / 10.40 |
| 0.5 | 96.77 / 7.53 | 96.36 / 8.71 |
| 1.5 | 96.58 / 7.41 | 96.79 / 8.69 |
| 1.9 | 96.38 / 7.55 | 96.81 / 9.12 |
| 1.95 | 94.88 / 8.75 | 94.19 / 11.00 |
| 1.0 (OURS) | **96.92 / 7.30** | **96.90 / 8.36** |

*Table 12.* Effect of centroid modeling on MLAAD (EER (↓)(%)) under Recipe-B training. A single multilingual model is trained on bona fide utterances from all eight languages. "Single-center" uses one global centroid, while "Multi-center" uses language-specific centroids.

| SETTING | FULL | DE | EN | ES | FR | IT | PL | RU | UK |
|---|---|---|---|---|---|---|---|---|---|
| SINGLE-CENTER | 28.61 | 35.02 | 23.36 | 43.32 | 30.13 | 34.56 | 20.67 | 41.23 | 20.50 |
| MULTI-CENTER | 17.70 | 24.45 | 12.07 | 29.20 | 24.75 | 17.80 | 5.14 | 19.94 | 15.07 |

# G. Additional Analysis for Multilingual Extension

This appendix reports an additional analysis of the multilingual extension on MLAAD under the strict real-only protocol. We focus on a single question: is language-conditioned (multi-center) centroid modeling necessary under cross-lingual shifts?

**Reference to the main method.** We follow the multi-domain formulation in Sec. 3.4. A single shared extractor $f_\theta$ is trained once on bona fide utterances from all languages. During training, language-conditioned centroids are updated online from bona fide embeddings only, and inference scores are computed by cosine similarity to the corresponding centroid when the language label is available (see Sec. 3.4).

**Language labels.** In MLAAD, we assume the test-time language label is available (e.g., via an off-the-shelf language identification system). This allows us to isolate the benefit of language-conditioned centroids under language-induced distribution shifts.

## G.1. Single-Center vs. Multi-Center Centroids on MLAAD

**Motivation.** Bonafide speech exhibits language-dependent shifts (e.g., phonetics, prosody, and recording characteristics), which can induce multi-modal structure in the embedding space. Under centroid-based one-class scoring, a single global centroid may therefore be overly restrictive.

**Experimental setup.** We train a **single multilingual model** on bona fide utterances from all eight MLAAD languages (en, de, es, fr, it, pl, ru, uk). We then compare two centroid modeling variants while keeping *the same extractor, training data, and objective*:

- **Single-center:** maintain one global centroid $c$ using bona fide embeddings pooled across all languages.

- **Multi-center:** maintain per-language centroids $\{c_\ell\}$ updated from bona fide embeddings of each language.

At test time, single-center uses $c$ for all trials, whereas multi-center uses the language-conditioned centroid $c_\ell$.

**Results.** Table 12 reports EER on the full MLAAD evaluation set and per-language subsets. Multi-center modeling substantially improves performance on the full set (from $28.61\%$ to $17.70\%$ EER) and consistently reduces EER across all eight languages.

**Discussion.** These results support the hypothesis that, under cross-lingual shifts, the bona fide embedding distribution is better represented by language-conditioned modes rather than a single global center. Consequently, language-conditioned centroids provide a more appropriate reference for strict one-class scoring in multilingual scenarios.

