# OpenReview forum: "Learning Tight Rejection Boundaries without Negatives for Strict One-Class Audio Deepfake Detection"
_ICML.cc/2026/Conference — ICML 2026 regular_

### Official Review · Reviewer_t29B · 2026-03-10

**Soundness:** 2
**Presentation:** 2
**Significance:** 2
**Originality:** 3
**Overall Recommendation:** 4
**Confidence:** 2

**Summary:**

This paper proposes CA-SOADD, a strict real-only one-class framework for audio deepfake detection without spoof supervision. It tightens the acceptance region by constructing off-manifold boundary probes and enforcing centroid compactness together with a centroid-referenced margin. It further extends to heterogeneous settings via domain-conditioned centroids and reports good generalization to unseen attacks and domain shifts.

**Compliance With Llm Reviewing Policy:**

Affirmed.

**Final Justification:**

Most of my concerns have been addressed, and I will raise my rating to WA.

**Key Questions For Authors:**

1. How robust is the proposed boundary tightening to truly unseen attacks, especially if the synthetic distribution-shifted views do not match real deepfake characteristics?

2. The method assumes the domain label is known at inference time. How realistic is this assumption in open-world deployment, and what happens when the domain label is unknown or incorrect?

3. Could the authors compare against more recent spoofed-speech detectors, strengthen the evidence-based related-work discussion, and report results on other attack types such as singing voice deepfakes?

**Limitations:**

The paper would be stronger with a clearer analysis of failure modes and sensitivity to probe design, and with evaluations when domain labels are unavailable or incorrect. The authors should also briefly discuss potential societal risks, including misuse and harms from false positives and false negatives in moderation or forensic use, and provide practical mitigation guidance such as responsible release and human oversight.

**Strengths And Weaknesses:**

Strengths:

1. The method avoids using spoof or auxiliary negative samples, and instead tightens the acceptance region via off-manifold boundary probes with a centroid-referenced margins.

2. The paper evaluates on standard benchmarks under unseen attack types and domain shifts, and includes ablations to support the contribution of the main components.

Weaknesses:

1. If the designed distribution-shifted views do not match the characteristics of real deepfakes, the tightened boundary may only be effective for these synthetic shifts and may not remain robust to truly unseen attacks.

2. The paper explicitly assumes that the domain label of each test utterance is known at inference time. Does this assumption still hold in an open-world deployment setting?

3. There is a lack of comparison with recent spoofed-speech detection methods. Under both the intrinsically real-only setting and the relaxed one-class setting evaluated in real-only mode, the chosen baselines are not sufficiently up to date.

4. The discussion of related work is insufficient, and many of the paper’s claims are not adequately supported by evidence. The authors should engage more thoroughly with recent literature and provide a deeper, better substantiated comparison between prior methods and the proposed approach.

5. How does the method perform under other types of audio spoofing attacks, such as singing voice deepfakes?

---

> ### Author Rebuttal · Authors · 2026-03-31
>
> Thanks for your careful review and thoughtful comments. We appreciate your positive feedback on the **real-only design, unseen-shift evaluation, and supporting ablations**. We hope the responses can well address your concerns.
>
> **Q1:** Robustness to truly unseen attacks when synthetic shifted views do not match real deepfake characteristics.
>
> **R1:** Thank you for this constructive comment. Robustness to truly unseen attacks remains a open challenge in OCL. Our approach addresses this by learning a more transferable rejection boundary without spoof supervision: synthetic shifted views serve as off-manifold boundary probes to tighten the bona fide acceptance region under cosine-to-centroid scoring, **rather than mimicking real deepfakes**. Our **benchmark-shift** and **spoof-family-shift** evaluation results confirm that the learned boundary **generalizes well to some unseen attacks differing substantially from the synthetic probes**, demonstrating that tightening the acceptance region itself drives generalization. Besides, we admit that we cannot fully solve the robustness problem, though broader probe families may further improve coverage. We will clarify this scope in the revision.
>
> **Q2:** The method assumes known domain labels at inference time. How realistic is this in open-world settings, and how does it perform when labels are missing or incorrect?
>
> **R2:** Thank you for this insightful comment.
>
> - In the paper, this assumption is used to **isolate the effect of domain-conditioned centroids under multilingual heterogeneity**, rather than to imply oracle labels are required in deployment.
> - In practice, language labels are often **available or can be reliably obtained** via an off-the-shelf language ID front-end.
> - For the **unknown-label case**, removing domain labels on MLAAD gives **18.07%** EER vs. **17.70%** with true labels, a mild drop.
> - For the **incorrect-label case**, performance degrades more noticeably: using fully random test-time labels yields **25.83%** EER, while a more realistic setting with **10% noisy domain labels** during training and testing gives **19.77%** EER.
>
> Overall, the method is **not critically dependent on domain labels**: it remains fairly robust when labels are unavailable and degrades gracefully under moderate label noise.
>
> **Q3:** Comparison with more recent spoofed-speech detectors, stronger related-work positioning, and evaluation on other attack types such as singing voice deepfakes.
>
> **R3:** Thank you for pointing it out. In the revision, we will make the following changes:
>
> - Expand the discussion to include **recent methods** such as KLOC (Speech Communication 2025) and OCKD (ICASSP 2024);
> - Clarify that our work follows a **strict real-only protocol** (no spoof data or labels in training, validation, or calibration), and that some recent detectors **are not directly comparable as strict-OCL baselines** due to their reliance on spoof information;
> - Extend **attack coverage** by adding an **extra experiment on CtrSVDD**, a controlled **singing-voice deepfake benchmark**. As illustrated in Table 3, our method achieves the **best overall performance**, indicating that the proposed boundary-shaping strategy **generalizes beyond speech-only spoofing**.
>
> **Table 2: Performance comparison (AUC / EER, %) on CtrSVDD.**
>
> | Model | AUC / EER |  | Model | AUC / EER |  | Model | AUC / EER |
> |:------|:----------|:--|:------|:----------|:--|:------|:----------|
> | LCNN | 85.5 / 21.9 |  | MPE | 83.3 / 23.5 |  | OC-SVM | 56.1 / 45.8 |
> | RawNet2 | 59.4 / 42.5 |  | ABCNET | 78.9 / 27.6 |  | DEEPSVDD | 65.3 / 39.3 |
> | RAWGAT | 81.2 / 27.1 |  | ASDG | 80.0 / 27.9 |  | OC-SOFTMAX | 52.4 / 48.7 |
> | AASIST | 84.5 / 21.8 |  | IG-SVD | 76.4 / 29.5 |  |  ACS | 56.4 / 45.5 |
> | **Ours** | **88.7 / 16.8** |  |  |  |  |  |  |
> ---
> **Q4:** Clearer analysis of failure modes, probe sensitivity, and robustness under unknown or incorrect domain labels.
>
> **R4:** Thank you for this important comment. We will add clearer analysis and revise the paper accordingly.
>
> **Failure modes:** We will make two failure modes more explicit: **cosine collapse** under strict real-only centroid training, which motivates Online Standardization, and **single-centroid miscalibration** under multilingual heterogeneity, which motivates domain-conditioned centroids on MLAAD.
>
> **Probe design:** We use one fixed probe pool $\mathcal{A}$ across benchmarks without dataset-specific tuning. Our probe-sensitivity study in Appendix C shows only moderate changes when individual probe types are removed, suggesting that the gain comes from the centroid-margin probe mechanism rather than any single operator.
>
> **Unknown domain labels & incorrectly specified labels:** This case is addressed in **Q2**.
>
> We will also expand the impact discussion to address misuse risks, false positives/negatives in moderation or forensic settings, and the need for responsible release, calibrated deployment, and human oversight in high-stakes use.

---

> > ### Author Rebuttal · Reviewer_t29B · 2026-04-03
> >
> > Thank you to the authors for their response to my concerns. The response to Q1 addresses part of my concern, and the benchmark-shift and spoof-family-shift results are helpful. However, they are still indirect with respect to the specific probe–attack mismatch question. I would still like to see how the method performs when the synthetic probes are more clearly mismatched with real unseen attacks. If stronger robustness requires introducing broader or more probe families, this may reduce the practical appeal of the method in real deployment.

---

> > > ### Author Response · Authors · 2026-04-05
> > >
> > > Thank you for the thoughtful follow-up. We appreciate the opportunity to further clarify the core philosophy of CA-SOADD, as our previous response may not have fully conveyed the underlying logic regarding the probe mechanism.
> > >
> > > The goal of CA-SOADD is to reduce dependence on spoof data, while learning from **structural generic forgery traces** (rather than specific known attack patterns) to improve robustness to **unseen attacks**.
> > >
> > > Our experimental setting follows prior works (e.g., **AAAI 2025[1], NDSS 2025[2], IEEE TDSC 2025[3]**), where models are trained on benchmark splits and evaluated on test sets containing unseen attacks. Our key distinction is that we remove **all spoof data** from both training and validation, thereby avoiding the cost of spoof collection.
> > >
> > > To more directly examine the reviewer's concern regarding **probe--attack mismatch**, we further trained CA-SOADD with reduced **2-probe** and **4-probe** subsets. As shown in Table 4, the performance degrades only **moderately**, and **Appendix C.5** shows a similar trend in the **5/6-probe sensitivity analysis**. These additional results indicate that the method does not rely too heavily on a large or closely matched probe set, and remains effective under increased mismatch.
> > >
> > > **Table 4: Robustness under reduced probe subsets for probe--attack mismatch analysis. Results are AUC ($\uparrow$) / EER ($\downarrow$) (%)**
> > >
> > > | Probe Combination | ASV2021-LA | ASV2021-DF | ASVspoof-5 |
> > > | :--- | :--- | :--- | :--- |
> > > | Perm + SpecMask + PhaseScr + IDINV | 96.24 / 8.06 | 95.88 / 9.03 | 92.25 / 13.72 |
> > > | PhaseScr + ReSamp + PNCR + VocHF | 95.98 / 8.18 | 96.27 / 8.86 | 92.43 / 13.90 |
> > > | PNCR + ReSamp + VocHF + IDINV | 95.89 / 8.51 | 95.72 / 9.11 | 91.86 / 13.86 |
> > > | Perm + ReSamp | 96.74 / 7.79 | 96.11 / 9.15 | 92.22 / 13.73 |
> > > | PNCR + VocHF | 95.90 / 8.85 | 95.65 / 10.08 | 91.05 / 15.42 |
> > > | Perm + IDINV | 95.56 / 8.94 | 95.82 / 9.86 | 91.94 / 14.06 |
> > > | **Ours** | **96.92 / 7.30** | **96.90 / 8.36**  | **92.74 / 13.44**|
> > > ---
> > >
> > > This behavior is consistent with our design motivation. The probes are **not intended for specific spoofing methods**. Instead, they are designed to capture **generic forgery traces** that are commonly observed in manipulated audio, including **temporal discontinuity, spectral information loss, phase inconsistency, bandwidth limitation, codec-induced reconstruction artifacts, vocoder-related high-frequency anomalies, and channel/compression perturbations**. For this reason, the probe pool is intentionally **redundant rather than specialized**. The performance in reduced-probe settings suggest that the learned decision boundary is not tightly coupled to particular known attack types. Instead, it captures generic forgery traces prevalent across diverse spoofing methods.
> > >
> > > As a result, for a unseen attack, the generated audio may still exhibit some of the seven generic characteristics summarized in our design, and can therefore still be captured by our probes. This may help explain why the method continues to perform well under attack mismatch. Consistent with this observation, as discussed in our response to the previous round’s **Q3**, our method also shows strong performance on the newly introduced **CtrSVDD** dataset.
> > >
> > > From a practical perspective, we believe an important advantage of CA-SOADD is its **weaker deployment assumption**. It can be trained and applied without spoof data, while still achieving stronger robustness to unseen attacks than **supervised** and **relaxed one-class baselines** in the standard setting.
> > >
> > > We hope that this clarification and the additional results may help address the reviewer’s concern. We are truly grateful for the reviewer’s thoughtful and constructive feedback.
> > >
> > > [1] Ren H, Lin L, Liu C H, et al. Improving generalization for ai-synthesized voice detection[C]//Proceedings of the AAAI Conference on Artificial Intelligence. 2025, 39(19): 20165-20173.
> > >
> > > [2] K. Kumari et al., “VoiceRadar: Voice Deepfake Detection using Micro-Frequency and Compositional Analysis,” in Proceedings 2025 Network and Distributed System Security Symposium, San Diego, 2025.
> > >
> > > [3] X. Li et al., “Critical Information Only: a Content Privacy-Preserving Framework for Detecting Audio Deepfakes,” IEEE Transactions on Dependable and Secure Computing, pp. 1–18, 2025.

---

### Official Review · Reviewer_oGbL · 2026-03-11

**Soundness:** 3
**Presentation:** 3
**Significance:** 3
**Originality:** 3
**Overall Recommendation:** 4
**Confidence:** 5

**Summary:**

This paper identifies three primary challenges in audio deepfake detection: first, detector overfitting to known attacks, which causes performance degradation; second, the lack of robust rejection boundaries in one-class learning; and third, the inadvertent introduction of new biases during model training. To address these issues, we propose a tri-objective learning method designed to establish tighter rejection boundaries through three core strategies: centroid compactness, benign invariance, and boundary shaping.

**Compliance With Llm Reviewing Policy:**

Affirmed.

**Key Questions For Authors:**

+ I’m not really sure of the motivation of the Geometry-Stabilized Representation Extractor, and why it works ?
+ The proposed method closely resembles standard contrastive learning by simulating positive and negative pairs for training. Consequently, the technical novelty appears limited, as the approach follows a well-established paradigm.
+ I am not entirely clear on the broader landscape of other one-class learning methods. Specifically, what are the major research directions in this field, and how does the CA-SOADD method contribute to the one-class learning community?

**Limitations:**

I have two more limitation suggestions. The paper only considers the ASVspoof dataset, however, recently have raised a series of new attacks, like SingFake, CodecFake, DFADD dataset. The paper didn’t validate the proposed loss function across different model backbones.

**Strengths And Weaknesses:**

The paper have following strengths:
+ The paper conducts comprehensive analysis on different datasets, such as ASVspoof-2021 (LA/DF) and ASVspoof-5, and also includes the MLADD.
+ It also has Tri-objective learning ablation study and embedding visualization.

I think the paper have several weaknesses can be improved:
+ The performance on MLADD (Table 2) suggests the model may not generalize well to cross-lingual data.
+ The effectiveness of the proposed loss function has not been validated across different backbones.

---

> ### Author Rebuttal · Authors · 2026-03-31
>
> Thank you for your careful review of our paper and thoughtful comments. We are encouraged by your positive comments on **our evaluation, ablation, and visual interpretability**. We hope the following responses can well address your concerns.
>
> **Q1:** Motivation and mechanism of the Geometry-Stabilized Representation Extractor?
>
> **R1:** Our method relies on **centroid-based cosine scoring under strict real-only training**, which requires a **geometrically stable embedding space**; otherwise, the running centroid and centroid-referenced margins become unreliable.
>
> - The extractor acts as a geometry-calibration module: layer/time aggregation reduces nuisance variation from the pretrained SSL encoder.
> - Bona fide-only Online Standardization (OS) aligns embeddings to a shared reference frame before centroid estimation and scoring. Appendix B provides direct evidence: without OS, embeddings show severe cosine collapse; with OS, angular dispersion increases substantially.
>
> We will clarify this in the revision.
>
> ---
>
> **Q2:** Novelty beyond standard contrastive learning and simulated positive/negative pairs?
>
> **R2:** We appreciate the question and want to clarify a potential misunderstanding. Our **objective is fundamentally different**.
>
> - Standard contrastive learning emphasizes pairwise or class-level separation, whereas our method learns a **boundary-centric one-class geometry**.
> - Bona fide samples are compacted around a shared centroid, while perturbed views are not semantic negatives but **off-manifold boundary probes** to **tighten the acceptance region** under the same cosine-to-centroid scoring used at inference.
>
> This distinction is crucial in strict OCL, where robustness to unseen attacks depends **more on learning a reliable rejection boundary than on separating against known negatives**. It is also supported by our **BCE / InfoNCE ablations**, which do not reproduce the **gains of the centroid-referenced margin**.
>
> ---
>
> **Q3:** Broader one-class learning landscape and CA-SOADD’s contribution to the OCL community?
>
> **R3:** Broadly, OCL includes boundary-based, reconstruction/density-based, and representation-centric methods. In audio anti-spoofing, **many recent “one-class” pipelines are effectively relaxed OCL**, still relying on spoof negatives or spoof-aware calibration. Our CA-SOADD contributes to strict centroid-based OCL by:
>
> - separating **compactness learning** from **boundary learning**.
> - enabling **real-only boundary tightening** using off-manifold probes without an explicit negative class.
> - preserving **score-objective consistency under cosine-to-centroid inference** while **mitigating cosine collapse**.
>
> ---
>
> **Q4:** Validation on newer attack benchmarks?
>
> **R4:** Due to time constraints, we report a supplementary result on CtrSVDD, a singing-voice benchmark requiring no background processing, where our method achieves the best performance (Table 2).
>
> **Table 2: Performance comparison (AUC / EER, %) on CtrSVDD.**
>
> | Model | AUC / EER | Model | AUC / EER | Model | AUC / EER |
> |:------|:----------|:------|:----------|:------|:----------|
> | LCNN | 85.5 / 21.9 | MPE | 83.3 / 23.5 | OC-SVM | 56.1 / 45.8 |
> | RawNet2 | 59.4 / 42.5 | ABCNET | 78.9 / 27.6 | DEEPSVDD | 65.3 / 39.3 |
> | RAWGAT | 81.2 / 27.1 | ASDG | 80.0 / 27.9 | OC-SOFTMAX | 52.4 / 48.7 |
> | AASIST | 84.5 / 21.8 | IG-SVD | 76.4 / 29.5 | ACS | 56.4 / 45.5 |
> | **Ours** | **88.7 / 16.8** | | | | |
> ---
> **Q5:** Cross-lingual generalization under MLAAD?
>
> **R5:** MLAAD is much more challenging than ASVspoof and serves as a **stress test** for **strict real-only detection under heterogeneous bona fide geometry**. Replacing a single global centroid with language-conditioned centroids **reduces MLAAD EER from 28.61% to 17.70%**, with **consistent gains across all eight languages**. We will clarify this point and note that, while the extension improves robustness under **cross-lingual shift**, there is still room for improvement.
>
> ---
>
> **Q6:** Validation of the proposed loss across different backbones?
>
> **R6:** We have validated the proposed objective across different SSL backbones by **replacing XLSR with WavLM and HuBERT**, keeping all other components **unchanged**. As shown in Table 3, the method remains effective in both cases, indicating that the gains are not tied to a specific encoder but arise from the **loss design and centroid-anchored boundary-shaping mechanism**. We will include these results in the revision.
>
> **Table 3: Backbone validation under the strict real-only one-class objective. Results are AUC (↑) / EER (↓) (%).**
>
> | Backbone | 2021 LA | 2021 DF | ASVspoof-5 | MLAAD |
> |:---|:---|:---|:---|:---|
> | WavLM | 95.62 / 10.51 | 92.67 / 14.55 | 88.18 / 16.38 | 83.09 / 21.53 |
> | Hubert | 94.98 / 11.08 | 91.31 / 15.35 | 84.73 / 19.90 | 85.62 / 19.34 |
> | XLSR | **96.92 / 7.30** | **96.90 / 8.36** | **92.74 / 13.44** | **86.49 / 17.70** |
> ---

---

> > ### Author Rebuttal · Reviewer_oGbL · 2026-04-02
> >
> > My concerns have been adequately addressed.

---

> > > ### Author Response · Authors · 2026-04-06
> > >
> > > Dear Reviewer oGbL,
> > >
> > > Thank you for your careful and constructive review. We sincerely appreciate your recognition of our empirical evaluation, ablations, and visualizations, as well as your valuable suggestions on the Geometry-Stabilized Representation Extractor, the distinction from contrastive learning, OCL positioning, and the need for broader validation.
> > >
> > > Your feedback has helped us clarify the role of the Geometry-Stabilized Representation Extractor, better articulate why CA-SOADD is a form of **boundary-centric one-class learning** rather than contrastive discrimination, and further strengthen the empirical support of the paper. In response, we added **cross-backbone validation with WavLM and HuBERT**, a supplementary evaluation on the **CtrSVDD singing-voice deepfake benchmark**, and a clearer discussion of the MLAAD results.
> > >
> > > **In addition, as a further consistency check**, we included a **reduced-probe subset analysis** in the second-round rebuttal (see **Table 4** in our response to Reviewer t29B). We hope this extra result provides additional support that the gains of our **off-manifold boundary-probe design** are not tightly tied to any single probe family, complementing the novelty evidence from the **BCE/InfoNCE ablations**.
> > >
> > > Thank you again for your rigorous and constructive evaluation. Your comments have been highly valuable in improving the clarity and overall quality of the paper.
> > >
> > > Best regards,
> > >
> > > Paper 7848 Authors

---

### Official Review · Reviewer_kptg · 2026-03-12

**Soundness:** 3
**Presentation:** 3
**Significance:** 3
**Originality:** 2
**Overall Recommendation:** 4
**Confidence:** 3

**Summary:**

This paper studies how to establish a tight rejection boundary around the bona fide distribution without relying on auxiliary negative samples for deep fake detection. The authors propose CA-SOADD, a framework utilizing a centroid-anchored tri-objective learning paradigm. It enforces centroid compactness, benign invariance, and employs distribution-shifted views as off-manifold boundary probes. The method is also extended to multi-domain scenarios using domain-conditioned centroids. Experiments on several benchmarks show that CA-SOADD outperforms other real-only baselines and maintains robustness against unseen attacks and distribution shifts.

**Compliance With Llm Reviewing Policy:**

Affirmed.

**Key Questions For Authors:**

1. Why does the benign invariance objective negatively impact performance in the multilingual (MLAAD) setting? Is there a theoretical or empirical explanation for why pulling benign augmentations closer to the clean representation interferes with language-specific centroid separation?
2. Could the authors clarify how the margin $m$ in `OBJ_cabs` is chosen, and how sensitive the overall performance is to this specific hyperparameter?

**Limitations:**

yes

**Strengths And Weaknesses:**

# Strengths

1. Soundness: The paper provides a very comprehensive evaluation, particularly highlighting zero-shot robustness across benchmarks (ASVSpoof-2021 to ASVSpoof-5) and specific spoof categories. The results convincingly show that the method prevents overfitting to seen spoof artifacts.
2. Presentation: The diagnosis of the "cosine-collapse" pathology and the online standardization solution is insightful.
3. Originality: Using structure-disrupting views merely as off-manifold boundary probes to enforce a centroid-referenced margin, instead of using them for surrogate binary classification, elegantly solves the objective-inference mismatch problem in strict one-class learning.

# Weaknesses

1. Significance: The reliance on a fixed set of structural probes (the $A$ operator pool) might still introduce some inductive bias, even if it is not spoof-specific.
2. Soundness: The extension to multi-domain (using domain-conditioned centroids) seems relatively straightforward compared to the core boundary-shaping contribution. Furthermore, it requires turning off the benign invariance objective on the MLAAD dataset, which slightly diminishes the universality of the proposed tri-objective framework.

---

> ### Author Rebuttal · Authors · 2026-03-31
>
> Thank you very much for your careful review of our paper and thoughtful comments. We are encouraged by your positive comments on **our zero-shot robustness, cosine-collapse diagnosis, and boundary-probe originality**. We hope the following responses could help clarify potential misunderstandings and alleviate your concerns.
>
> **Q1:** Why does the benign invariance objective negatively impact performance in the multilingual (MLAAD) setting? Is there a theoretical or empirical explanation for why pulling benign augmentations closer to the clean representation interferes with language-specific centroid separation?
>
> **R1:** Thank you for your insightful question. We view this as a **domain-dependent trade-off** rather than a limitation of the framework.
>
> - OBJ$_{\text{binv}}$ is effective in **relatively homogeneous settings**, where it stabilizes the **local centroid neighborhood**. In multilingual MLAAD, it can **suppress language-discriminative variation**, leading to **reduced separation among language-conditioned centroids and weaker cosine-to-centroid scoring**.
> - We will clarify this as a trade-off between **intra-domain compactness** and **cross-domain centroid geometry**, and note that better integration with **heterogeneous multi-centroid structure** is an important direction for future work.
>
> ---
>
> **Q2:** Could the authors clarify how the margin $m$ in $OBJ_{cabs}$ is chosen, and how sensitive the overall performance is to this specific hyperparameter?
>
> **R2:** Thank you for your insightful comment.
>
> - In OBJ$_{\text{cabs}}$, $m$ controls the **required similarity gap** between a bona fide embedding and its off-manifold probe relative to the centroid, i.e., the strength of boundary tightening under the same cosine-to-centroid geometry used at inference.
> - We choose **$m=1$ as a moderate default** under the **strict real-only protocol**. If $m$ is too small, the boundary-shaping signal is weak; if it is too large, optimization becomes overly restrictive.
> - To assess sensitivity, we have conducted a rebuttal-time ablation on ASVspoof 2021. As shown in Table 1, performance **remains fairly stable for $m \in [0.5,1.9]$**, with the best results at $m=1$. When $m$ becomes too large, performance degrades on both LA and DF, consistent with **over-tightening**.
>
> We will include this table in the revision and clarify that the method is **moderately sensitive but not fragile** around the default choice.
>
> **Table 1: Margin $m$ ablation under the strict real-only protocol. Results are AUC (↑) / EER (↓) (%).**
>
> | margin $m$ | 2021 LA | 2021 DF |
> |:---|:---|:---|
> | 0.1 | 96.69 / 7.44 | 94.85 / 10.40 |
> | 0.5 | 96.77 / 7.53 | 96.36 / 8.71 |
> | 1 (Ours) | **96.92 / 7.30** | **96.90 / 8.36** |
> | 1.5 | 96.58 / 7.41 | 96.79 / 8.69 |
> | 1.9 | 96.38 / 7.55 | 96.81 / 9.12 |
> | 1.95 | 94.88 / 8.75 | 94.19 / 11.00 |
> ---
>
> **Q3:** Does the fixed operator pool $\mathcal{A}$ introduce an inductive bias that compromises the method's generic nature?
>
> **R3:** We agree that a fixed probe pool may introduce some inductive bias. However, this bias is **generic rather than spoof-specific**: $\mathcal{A}$ does not use spoof data, remains fixed across benchmarks, and is not tuned to particular attacks. Its role is simply to provide off-manifold probes for boundary shaping.
>
> Our claim is therefore not that $\mathcal{A}$ is **bias-free**, but that boundary tightening can be achieved **without spoof samples, labels, or surrogate negatives**. In the previous manuscript, we used a unified fixed pool to isolate this mechanism under a **strict real-only protocol**. We will clarify this limitation and note that $\mathcal{A}$ can be **extended with broader generic perturbations in future work**.
>
> ---
>
> **Q4:** Regarding the novelty of the multi-domain extension and the potential impact on the framework's universality when disabling objectives on MLAAD.
>
> **R4:** We agree that the multi-domain extension is lighter than the core boundary-shaping contribution, and we will revise the paper to make this hierarchy clearer. Our **main contribution** is the **centroid-anchored boundary-shaping mechanism under strict real-only learning**, while the domain-conditioned centroid design is a **deployment-oriented extension** that illustrates how this mechanism can be instantiated when bona fide speech is heterogeneous rather than approximately unimodal.

---

### Decision · Program_Chairs · 2026-04-30

**Decision:**

Accept (regular)

**Comment:**

This paper proposes a strict real-only one-class framework (CA-SOADD) for audio deepfake detection that learns a tight rejection boundary via centroid-anchored compactness, consistency, and boundary shaping with off-manifold probes. Reviewers generally found the problem important and the approach technically sound, highlighting strong empirical results, robustness to unseen attacks, and a boundary-learning mechanism. After the rebuttal, most reviewers (all “Weak Accept”) acknowledged that their concerns were largely addressed and supported acceptance. Some concerns remain regarding potential inductive bias from probe design, limited evaluation on broader settings, and clarity in positioning and novelty. However, I think these issues are relatively minor and have been reasonably addressed by additional experiments and clarifications.